# M1BP cooperates with CP190 to activate transcription at TAD borders and promote chromatin insulator activity

Indira Bag [1,2], Shue Chen [1,2,4], Leah F. Rosin [1,2,4], Yang Chen [1,2], Chen-Yu Liu[3], Guo-Yun Yu[3] & Elissa P. Lei [1,2 ✉]

Genome organization is driven by forces affecting transcriptional state, but the relationship between transcription and genome architecture remains unclear. Here, we identified the *Drosophila* transcription factor Motif 1 Binding Protein (M1BP) in physical association with the *gypsy* chromatin insulator core complex, including the universal insulator protein CP190. M1BP is required for enhancer-blocking and barrier activities of the *gypsy* insulator as well as its proper nuclear localization. Genome-wide, M1BP specifically colocalizes with CP190 at Motif 1-containing promoters, which are enriched at topologically associating domain (TAD) borders. M1BP facilitates CP190 chromatin binding at many shared sites and vice versa. Both factors promote Motif 1-dependent gene expression and transcription near TAD borders genome-wide. Finally, loss of M1BP reduces chromatin accessibility and increases both inter- and intra-TAD local genome compaction. Our results reveal physical and functional interaction between CP190 and M1BP to activate transcription at TAD borders and mediate chromatin insulator-dependent genome organization.

[1] Nuclear Organization and Gene Expression Section, Bethesda, MD, USA. [2] Laboratory of Biochemistry and Genetics, Bethesda, MD, USA. [3] Laboratory of Cellular and Developmental Biology, National Institute of Diabetes and Digestive and Kidney Diseases, National Institutes of Health, Bethesda, MD, USA. [4]These authors contributed equally: Shue Chen, Leah F. Rosin. ✉email: leielissa@niddk.nih.gov

I n eukaryotic cells, the three-dimensional organization of the genome plays a critical role in achieving proper spatial and temporal patterns of gene expression during development. Chromatin insulators are DNA-protein complexes involved in the establishment, maintenance, and regulation of nuclear organization to modulate gene expression (reviewed in[1,2]). Insulators regulate interactions between cis-regulatory elements such as enhancers and promoters and demarcate silent and active chromatin regions to ensure their proper regulation. They can inhibit the interaction between an enhancer and a promoter when positioned between the two elements and can act as a barrier to stop repressive chromatin from spreading over active genes. Furthermore, chromatin insulators can promote intra- and inter-chromosomal looping to control topology of the genome. Certain insulator proteins are highly enriched at the self-interacting boundaries of topologically associating domains (TADs) throughout the genome. In mammals, only a single insulator protein, CCCTC-binding Factor (CTCF), has thus far been identified, and CTCF indeed is enriched at TAD borders and is required for TAD formation. In contrast, *Drosophila melanogaster* CTCF is not particularly enriched at TAD borders, and a recent study indicates that CTCF plays a limited role in TAD formation in flies[3]. In fact, *Drosophila* harbors a variety of insulator protein complexes, all of which contain the protein Centrosomal protein 190 (CP190). CP190 is highly enriched at TAD borders, suggesting a possible role in TAD formation. Another notable feature of genome organization that has been explored in detail in *Drosophila* is the key role of transcription and the presence of constitutively active genes at TAD borders[4–6]. General inhibition of transcription using chemical treatments or heat shock results in disruption of TADs and compartments, but the mechanistic details of how transcription contributes to genome organization are yet to be elucidated.

The *Drosophila gypsy* insulator, also known as the Suppressor of Hairy wing [Su(Hw)] insulator, was the first characterized CP190-containing insulator complex. The zinc-finger DNA-binding protein Su(Hw) provides binding specificity of the complex, and both CP190 and the Modifier of mdg4 [Mod(mdg4)] 67.2 kDa isoform [Mod(mdg4)67.2] contain an N-terminal Broad-Complex, Tramtrack, and Bric a brac (BTB) domain that can homo-dimerize or heterodimerize to facilitate insulator–insulator interactions and promote formation of long range insulator-mediated loops[7–10]. Initially, the *gypsy* insulator complex was characterized as binding the 5'-untranslated region of the *gypsy* retroelement. However, the core complex also binds thousands of endogenous sites throughout the genome and can function similarly at least at a subset of those sites[11–15]. Moreover, the three *gypsy* insulator core components do not colocalize absolutely at all binding sites throughout the genome, and each protein can interact with other insulator proteins[13,16–19]. In diploid interphase nuclei, *gypsy* insulator proteins coalesce into large foci termed insulator bodies. These structures can be induced by stress[20], and insulator bodies have also been proposed to serve as storage depots for insulator proteins[21,22]. Nevertheless, there is a high correlation between proper insulator function and insulator body localization[7,10–16,23–27]. In summary, the *gypsy* insulator complex contributes to higher order nuclear organization on several levels.

CP190 also associates with a variety of additional DNA-binding proteins that likely impart specificity of the respective complex. The BED finger-containing proteins BEAF-32, Ibf1, and Ibf2 interact with CP190 and promote insulator function[17]. Three additional zinc-finger proteins Pita, ZIPIC, and CTCF also interact with CP190 and contribute to insulator activity[18,28]. Recently, the zinc-finger protein CLAMP was demonstrated to positively affect *gypsy* insulator activity and to colocalize particularly with CP190 at promoters throughout the genome[16]. In addition, previous work showed that CP190 preferentially binds Motif 1-containing promoters[29], but the functional significance of this observation is currently unknown. The precise functions of CP190, its associated factors, as well as their relationship with transcription regulation have not yet been elucidated.

Motif 1 binding protein (M1BP) is a ubiquitously expressed transcriptional activator that is required for the expression of predominantly constitutive genes. A zinc-finger DNA-binding protein, M1BP, specifically binds to the core promoter element Motif 1 consensus sequence that is distinct from the canonical TATA box and mainly controls the expression of constitutively active genes that are transiently paused[30]. For example, M1BP interacts with the TATA-binding protein-related factor 2 (TRF2) to activate transcription of ribosomal protein genes in a Motif 1-dependent manner[31]. Finally, recent studies found that Motif 1 and M1BP are highly enriched at TAD boundaries along with CP190 and BEAF-32[32–35]. Depletion of M1BP led to increased inter-chromosomal Hi-C contacts; however, concomitant cell cycle disruption precluded interpretation of these results[33]. The possible role of M1BP-dependent transcriptional regulation in genome organization has not yet been interrogated in detail.

In this study, we identify M1BP as a physical interactor and positive regulator of the *gypsy* insulator complex. Depletion of M1BP decreases *gypsy*-dependent enhancer blocking and barrier activities and reduces the association of the core insulator complex with the *gypsy* insulator sequence. ChIP-seq analysis reveals extensive genome-wide overlap of M1BP particularly with promoter-bound CP190, and depletion of M1BP results in extensive loss of CP190 chromatin association genome-wide. Depletion of CP190 also disrupts M1BP binding at many of its binding sites. Nascent euRNA-seq (neuRNA-seq) analysis of M1BP- or CP190-depleted cells indicates that both factors co-regulate a similar set of genes genome-wide. In particular, loss of gene activation correlates with disrupted M1BP and CP190 binding, and these events are frequently observed at TAD borders. Depletion of M1BP disrupts *gypsy* insulator body localization within the nucleus and alters both inter- and intra-TAD local genome compaction. Finally, knockdown of M1BP decreases chromatin accessibility at its binding sites, including genes that it activates and regions in proximity of TAD borders. Taken together, our findings identify a mechanistic relationship between M1BP and CP190 to activate Motif 1-dependent transcription as well as to promote chromatin insulator activity and nuclear organization.

## Results

**M1BP interacts physically with core *gypsy* insulator proteins**. In order to identify additional *gypsy* insulator interactors, we performed immunoaffinity purification of *Drosophila* embryonic nuclear extracts using antibodies specific for Su(Hw) or CP190 and analyzed the eluates using quantitative mass spectrometry. In addition to the third *gypsy* core component Mod(mdg4)67.2 and previously reported interactors such as Ibf1, Ibf2, BEAF-32, and CTCF, we identified M1BP in both purifications (Supplementary Tables 1 and 2). CP190 was also previously identified as a physical interactor by co-immunoprecipitation of M1BP by mass-spec analysis using *Drosophila* embryonic nuclear extract and subsequently confirmed by western blotting[36]. We verified that all three *gypsy* core components interact with M1BP by performing anti-Su(Hw), anti-CP190, and anti-Mod(mdg4)67.2 purifications and western blotting (Fig. 1a–c). Finally, we verified that Su(Hw), CP190, and Mod(mdg4)67.2 could be immunopurified by anti-M1BP but not normal serum (Fig. 1d). In contrast, an unrelated factor, Polycomb, was not co-immunoprecipitated with any of these antibodies. These results indicate that M1BP is associated

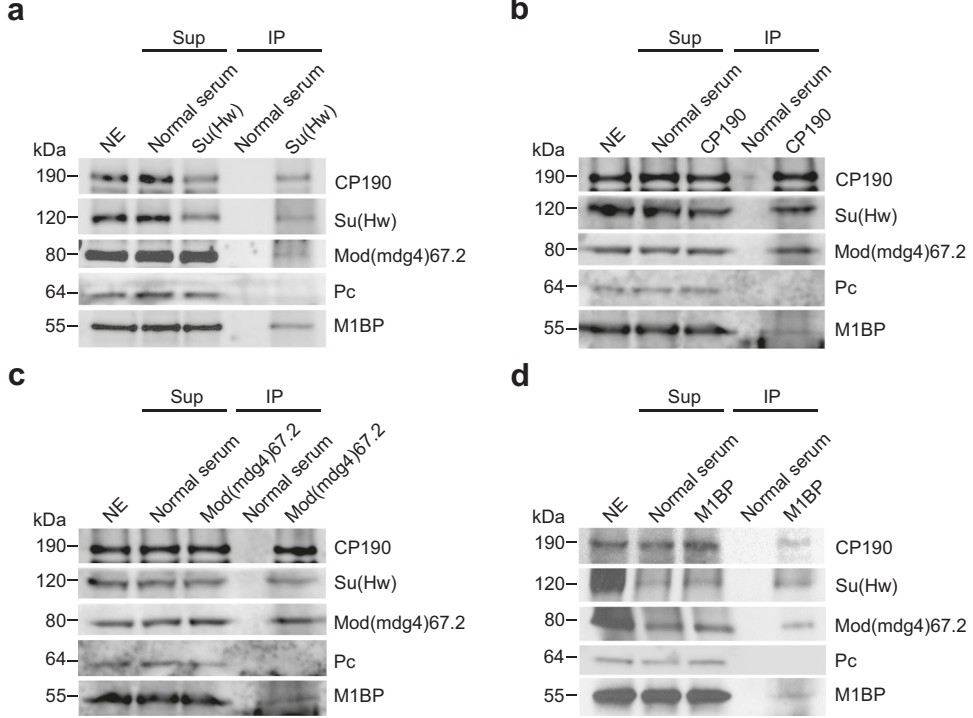

**Fig. 1 Co-immunoprecipitation of core *gypsy* components with M1BP. a** Nuclear extracts (NE) from embryos aged 0–24 h were immunoprecipitated with either normal guinea pig serum or guinea pig anti-Su(Hw) antibody. Unbound supernatant (Sup) and bound (IP) fractions are shown. Polycomb (Pc) is shown as a negative control. In all, 10% of NE is loaded in the gel. We calculated that 5.2% of total CP190, 7.2% of Su(Hw), 1.9% of Mod(mdg4)67.2, and 5.3% of M1BP were immunoprecipitated with anti-Su(Hw) antibody. **b** Immunoprecipitation of M1BP with normal guinea pig serum or guinea pig anti-CP190 is shown. Overall, 12% of total CP190, 7.0% of Su(Hw), 8.2% of Mod(mdg4)67.2, and 3.8% of M1BP were immunoprecipitated with anti-CP190 antibody. **c** Immunoprecipitation of M1BP with normal rabbit serum or rabbit anti-Mod(mdg4)67.2 is shown. Overall, 11% of total CP190, 8.5% of Su(Hw), 10% of Mod(mdg4)67.2, and 3.0% of M1BP were immunoprecipitated with anti-Mod(mdg4)67.2 antibody. **d** Immunoprecipitation of M1BP with normal rabbit serum or rabbit anti-M1BP is shown. Overall, 3.8% of total CP190, 2.6% of Su(Hw), 1.4% of Mod(mdg4)67.2, and 3.7% of M1BP were immunoprecipitated with anti-M1BP antibody. Samples from the same experiment were run on different gels for proteins with similar molecular weights. All western blotting experiments were performed with at least two independent biological replicates for each antibody, and a single experiment is shown.

physically either directly or indirectly with the core *gypsy* insulator proteins.

**M1BP promotes *gypsy* insulator function.** Given their physical interaction, we sought to test whether M1BP may play a role in *gypsy* insulator function. To deplete M1BP levels in vivo, we utilized an *M1BP^RNAi* fly line expressing a hairpin under upstream activating sequence (UAS) control that generates siRNAs against *M1BP* when combined with a Gal4 driver. To validate the knockdown efficiency of this line, we performed western blot analysis of larval extracts from flies expressing this RNAi construct using the ubiquitously expressed *Act5C-Gal4* driver compared to a control line expressing driver alone (Fig. 2a). Importantly, depletion of M1BP does not have any effect on overall Su(Hw), CP190 or Mod(mdg4)67.2 protein levels. Using *Act5C-Gal4*, *M1BP^RNAi* resulted in 100% late larval lethality, and complete pupal lethality is observed using the muscle-specific *Mef2-Gal4* driver (Supplementary Table 3). With CNS-enriched *l(3)31-1-Gal4* or wing-expressed *Ser-Gal4*, *M1BP^RNAi* expression, flies remain completely viable.

In order to investigate whether M1BP affects *gypsy*-dependent enhancer-blocking activity, we examined the effect of M1BP depletion on the well-characterized allele *ct⁶*. This loss-of-function allele results from *gypsy* retrotransposon insertion between the promoter and distal wing margin enhancer of *cut* (Fig. 2b). The *gypsy* insulator blocks communication between the two elements, reducing *cut* expression and causing disruption of the wing margin visible in the adult fly[37]. We used a scoring scale

from 0 to 4 with increasing severity of wing margin notching corresponding to higher insulator activity. We found that knockdown of *M1BP* driven by *Ser-Gal4* compared to driver alone restored wing margin tissue, consistent with a decrease in enhancer-blocking activity (Fig. 2c). We also determined that knockdown of *M1BP* does not result in changes in wing margin induced by the *gypsy*-independent *ctⁿ* loss-of-function allele, which is caused by insertion of the *roo* transposable element (Supplementary Fig. 1). These results indicate that M1BP is required for *gypsy*-dependent enhancer blocking activity.

We next investigated the effect of M1BP depletion on *gypsy*-dependent barrier activity in a variety of tissues. In this assay, we used a *UAS-luciferase* reporter that is either insulated by flanking Su(Hw)-binding sites or not insulated, with either reporter inserted into the same genomic site (Fig. 2d)[24]. We performed this quantitative luciferase-based assay using different tissue-specific Gal4 drivers to control both the luciferase reporter and *M1BP^RNAi*. Ubiquitously expressed *Act5C-Gal4* promotes high luciferase expression in insulated compared to non-insulated control larvae (Fig. 2e and Supplementary Table 4). As a positive control, knockdown of *su(Hw)* resulted in dramatic decrease in luciferase expression only in the insulated line, suggesting loss of barrier function. Likewise, knockdown of *M1BP* driven by *Act5C-Gal4* resulted in statistically significant reduction in luciferase activity compared to the insulated control line. Furthermore, knockdown using either the muscle-specific *Mef2-Gal4* driver or the CNS-enriched *l(3)31-1-Gal4* driver also resulted in significant decreases in luciferase activity (Fig. 2f, g and Supplementary

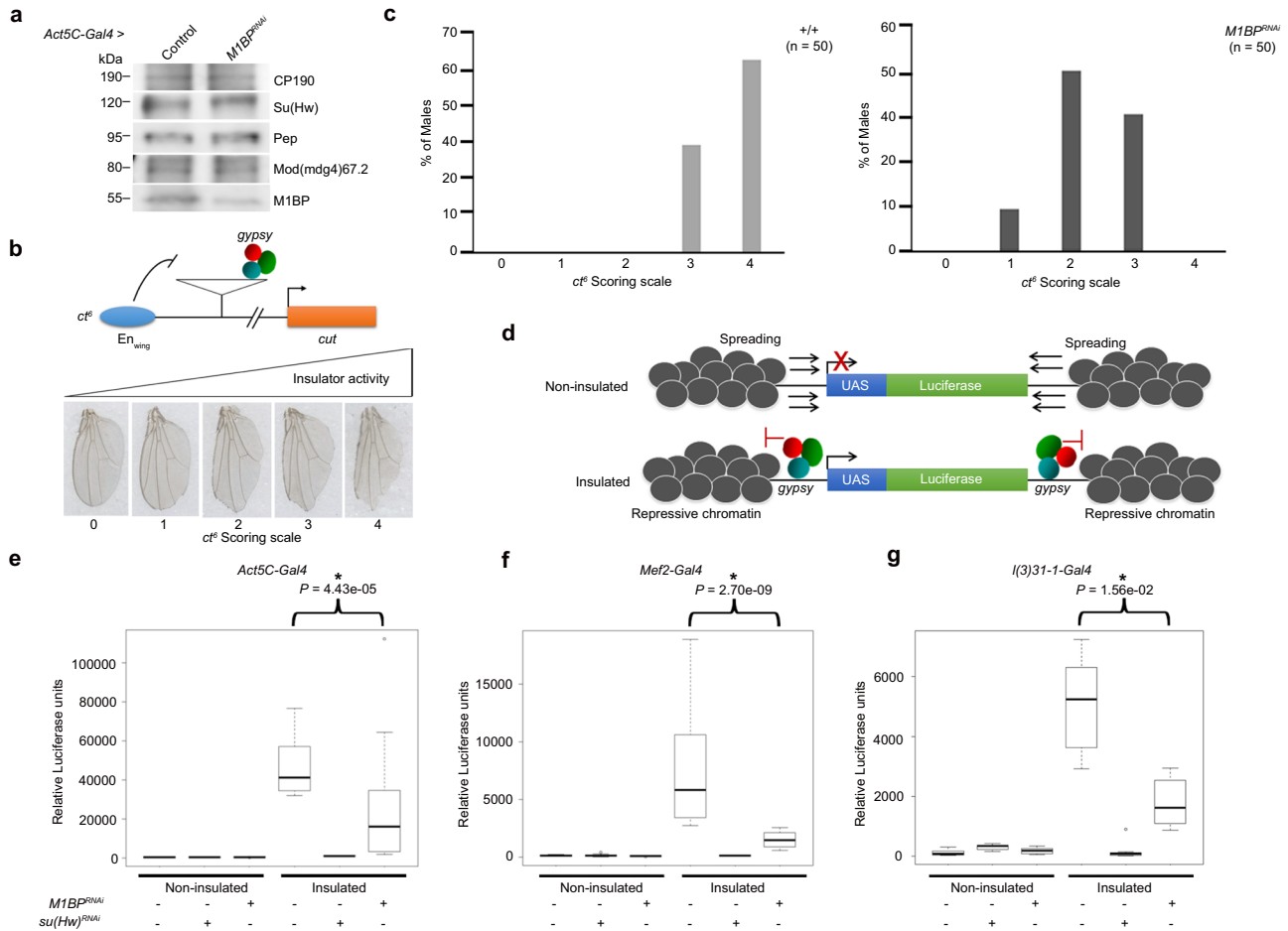

**Fig. 2 M1BP promotes *gypsy*-dependent enhancer-blocking and barrier activities. a** Western blotting of male third instar larval extracts for M1BP, insulator proteins, and Pep loading control in control and *M1BP^RNAi* knockdown flies using *Act5C-Gal4* driver. Samples from the same experiment were run on different lanes of the same gel for proteins with similar molecular weights. Western blotting experiments were performed using two independent biological replicates with similar results. **b** M1BP promotes enhancer-blocking activity at *ct^6*. Top: schematic diagram of reporter system. The *gypsy* retrotransposon is inserted in between the promoter of *cut* and wing margin (En_Wing) enhancer. Bottom: insulator activity for *ct^6* was scored in male flies on a scale of 0–4. 0, no notching; 1, slight notching in anterior tip of wing; 2, mild notching throughout posterior wing; 3, extensive notching both in anterior and posterior wing; 4, severe notching throughout the anterior and posterior wing. **c** Graph represents quantification of *ct^6* wing phenotype of male wild-type (+/+) and *M1BP^RNAi* using *Ser-Gal4* driver. *n*, total number of flies scored. **d** Depletion of M1BP shows reduced *gypsy*-dependent barrier activity in all tissues tested. Schematic diagram of non-insulated *UAS-luciferase* system shows spreading of repressive chromatin can reduce luciferase expression, but presence of the *gypsy* insulator acts as a barrier and allows for luciferase activity. **e** Relative luciferase activity of insulated or non-insulated male larvae of control and *M1BP^RNAi* driven by *Act5C-Gal4* driver, **f** *Mef2-Gal4* driver, and **g** *l(3)31-1-Gal4* driver. Luciferase values of all genotypes are plotted as box and whisker plots using one-way ANOVA followed by Tukey HSD post hoc tests to calculate *P* values for pairwise comparisons. The box represents the 25–75th percentiles, and the median is indicated. The whiskers show the minimum and maximum values. For each genotype, *n* = 12 individual larvae. Bracket indicates *P* values of two-way comparisons (**P* < 0.05) and all comparisons are shown in Supplementary Table 4.

Table 4). We conclude that M1BP promotes *gypsy*-dependent barrier activity in all tissues tested.

**M1BP extensively colocalizes with CP190 genome-wide.** In order to obtain high-resolution information about the genome-wide chromatin association of M1BP and its relationship with *gypsy* core components, we performed ChIP-seq analysis for M1BP, CP190, Su(Hw), and Mod(mdg4)67.2 in the embryonic Kc167 (Kc) hemocyte cell line. Using previously validated antibodies, we identified 3121 M1BP, 8022 CP190, 4638 Su(Hw), and 3536 Mod(mdg4)67.2 peaks. By western blot analysis, we validated efficient protein depletion of M1BP and no effect on insulator protein levels 5 days after *M1BP* double stranded RNA (dsRNA) transfection (Fig. 3a and Supplementary Fig. 2a). Similar to previous work[30], we also verified that this level of

M1BP depletion did not greatly alter cell viability or lead to accumulation of cells in M-phase (Supplementary Fig. 2b, d–e), in contrast to a previous study that used more dsRNA for a longer time period[33]. Our ChIP-seq analyses confirm that depletion of M1BP by RNAi dramatically reduced M1BP binding to chromatin throughout the genome, also confirming the specificity of the antibody (Fig. 3b). Interestingly, we found that 79% (2461) of total M1BP peaks overlap with CP190 genome-wide (31% of total CP190 peaks) (Fig. 3b, c). In contrast, we found very low overlap of M1BP with either Su(Hw) (5.2%) or Mod(mdg4)67.2 (7.0%) (Fig. 3c). Finally, M1BP also overlaps considerably with BEAF-32 (41%) but less substantially with CLAMP, ZIPIC, Pita, Ibf1, Ibf2, and CTCF (Supplementary Fig. 3).

We further examined the distribution of M1BP, CP190, and shared M1BP-CP190 sites with respect to genomic features. Consistent with earlier reports, we verified that both M1BP and

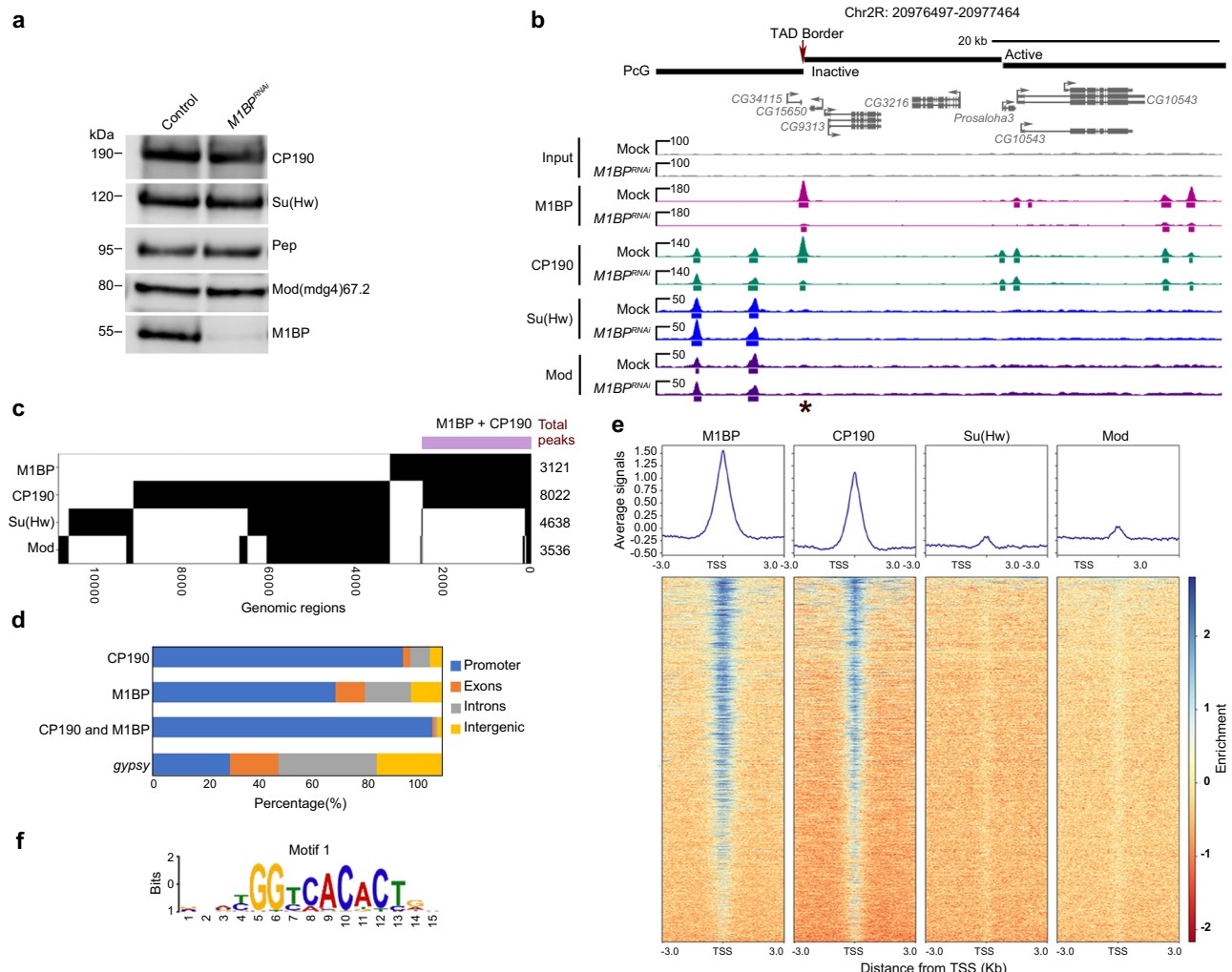

**Fig. 3 M1BP extensively colocalizes specifically with CP190 throughout the genome. a** Western blotting of total lysates from Kc control and *M1BP* knockdown cells showing knockdown efficiency of *M1BP* and no effects on protein levels of insulator proteins, with Pep as loading control. Samples from the same experiment were run on different lanes of the same gel for proteins with similar molecular weights. Western blotting was performed for three independent biological replicates with similar results. **b** Screenshot example of ChIP-seq profiles showing in Kc cells M1BP co-localizing with CP190 but not Su(Hw) or Mod(mdg4)67.2. Asterisk indicates a particular peak of interest measured in Fig. 6c, site 7. **c** Binary heatmap of M1BP, CP190, Mod(mdg4)67.2, and Su(Hw) binding sites in control cells ordered by supervised hierarchical clustering. Each row represents a single independent genomic location, and a black mark in a column represents the presence of a particular factor: 2461 (79%, purple bar) peaks of 3121 total M1BP peaks overlap with CP190, 163 (5.2%) M1BP peaks overlap with Su(Hw) peaks, and 217 (7%) M1BP peaks overlap with Mod(mdg4)67.2. **d** Bar plot shows distribution of M1BP, CP190, and *gypsy* (Su(Hw)/Mod(mdg4)67.2/CP190 all present) binding sites with respect to genomic features. **e** Heatmaps of M1BP, CP190, Su(Hw), and Mod (mdg4)67.2 peaks in control cells sorted independently by decreasing average ChIP-seq signals normalized to inputs. Reads are centered on 6307 TSSs containing Motif 1 site. The horizontal axis corresponds to distance from TSS with Motif 1 site. **f** Motif 1 consensus sequence.

CP190 binding, but not Su(Hw) or Mod(mdg4)67.2 binding, are enriched at transcription start sites (TSS) (Fig. 3e). As expected, we verified that these promoters frequently harbor Motif 1 consensus sequence[30,38] (Fig. 3f and Supplementary Data, 54% of M1BP, 35% of CP190, and 54% co-occupied promoter peaks). Therefore, our ChIP-seq results indicate that M1BP colocalizes primarily with CP190 throughout the genome, particularly at Motif 1-containing promoters.

**M1BP and CP190 regulate transcription of a similar gene set.** Given that M1BP and CP190 colocalize extensively at promoters enriched for the presence of Motif 1, we compared how each factor affects transcription genome-wide. We performed neuRNA-seq after a 1 h pulse labeling of Kc cells to examine newly synthesized transcripts in mock transfected control cells

versus knockdown of *M1BP*, *Cp190*, or *mod(mdg4)* treatment specific for the 67.2 KDa isoform. We found that depletion of M1BP results in upregulation of nascent transcription of 1315 genes and downregulation of 607 genes (Fig. 4a). Importantly, we found that only 22% of promoters of upregulated genes (Fisher's exact tests [FET], $P = 1.6e{-}02$, odds ratio = 0.8) but 46% of promoters of downregulated genes harbor M1BP chromatin binding based on ChIP-seq analysis in the control condition (FET, $P < 2.2e{-}16$, odds ratio = 2.8). These results are consistent with an earlier report that M1BP is mainly directly involved in transcriptional activation[30]. Likewise, CP190 binds promoters of only 40% of genes that are upregulated after M1BP knockdown (FET, $P = 7.3e{-}02$, odds ratio = 0.9), but CP190 binding is highly enriched at gene promoters that are downregulated (76%, FET, $P < 2.2e{-}16$, odds ratio = 4.4) resulting from M1BP depletion. Taken together, these results suggest that CP190 may cooperate

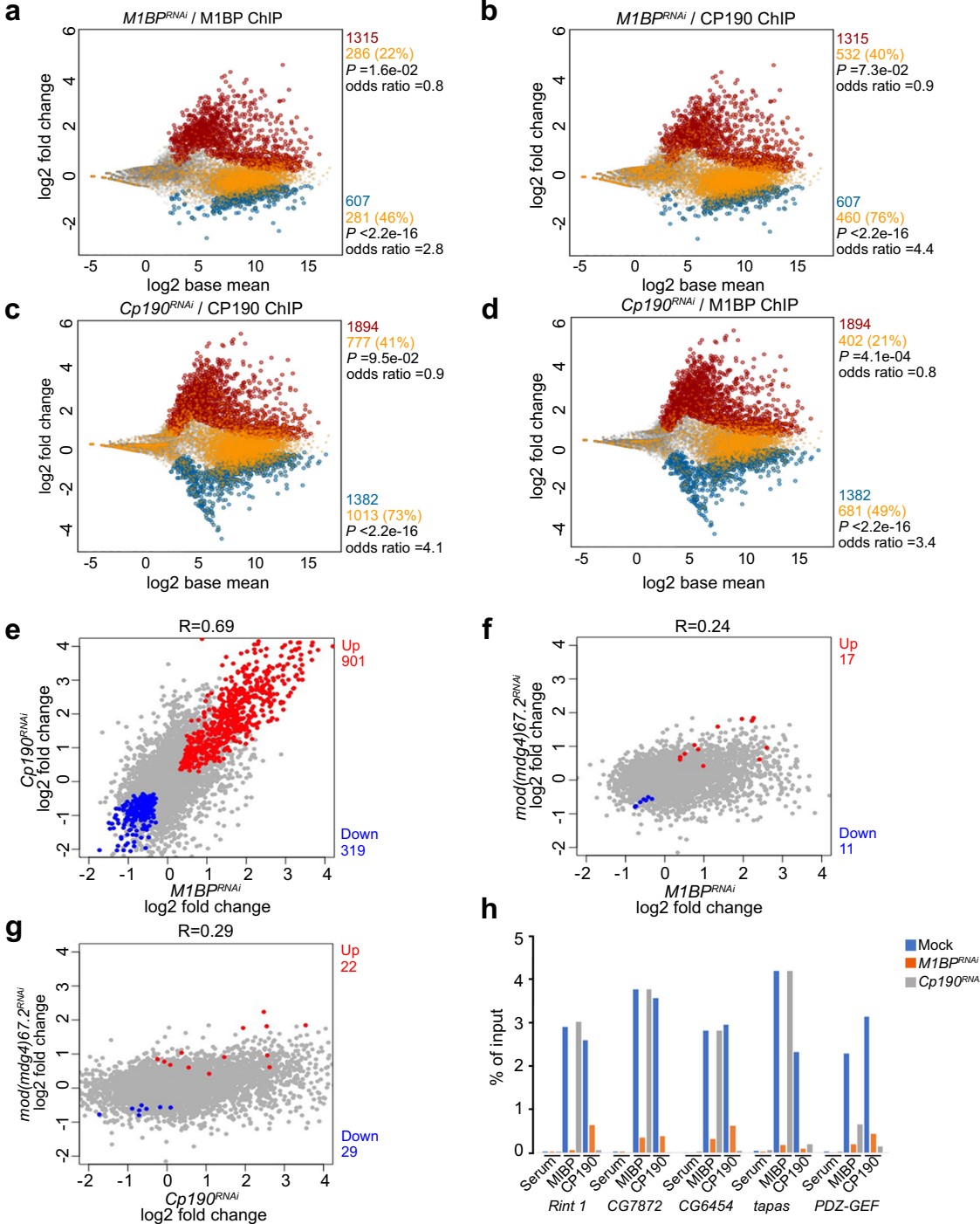

**Fig. 4 M1BP and CP190 transcriptionally regulate a common set of genes. a**, **b** MA plots showing changes in neuRNA levels upon depletion of M1BP. Statistically significant changes include 1315 upregulated genes (red) and 607 downregulated genes (blue) using $P_{adj} < 0.05$. Unchanged genes are indicated in gray. Gene promoters containing M1BP peaks (**a**) or CP190 peaks (**b**) are additionally colored yellow. Two-sided Fisher's exact test was used. **c**, **d** MA plots showing neuRNA-seq affected genes after depletion of CP190. Significantly upregulated genes (1894, red) and downregulated genes (1382, blue) are shown. Genes containing CP190 (**c**) or M1BP (**d**) at their promoters are shown in yellow. Fisher's exact test (two-sided) was used. **e**–**g** Scatter plots comparing neuRNA-seq profiles of *M1BP* and *Cp190* knockdowns (**e**), *M1BP* and *mod(mdg4)* knockdowns (**f**), or *Cp190* and *mod(mdg4)* knockdowns (**g**). Pearson's R corresponds to correlation coefficient between two profiles. Common upregulated genes are indicated in red, and common downregulated genes are indicated in blue. **h** Differentially decreased CP190 peaks in *M1BP^RNAi^* associated with the promoter of downregulated genes affected in both *M1BP* and *Cp190* knockdowns are verified by ChIP-qPCR. Percent input DNA precipitated is shown for each primer set. Average values from $n = 2$ biological replicates measured using four technical replicates are shown. Detailed description of each site is summarized in Supplementary Table 5, and $C_t$ values are listed in Source Data 7.

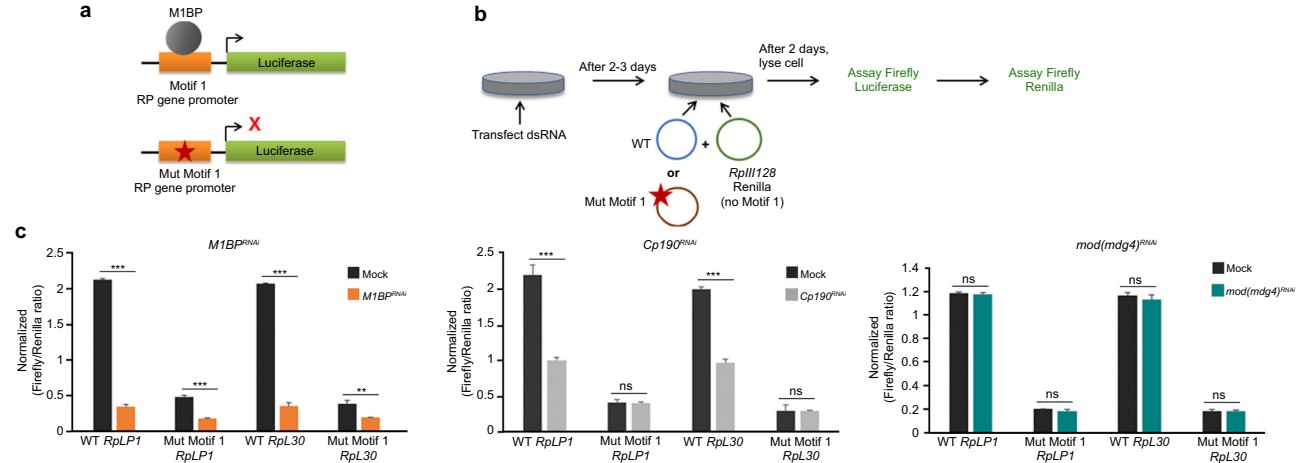

**Fig. 5 M1BP and CP190 both activate Motif 1-containing gene promoters. a** Schematic diagram of plasmids containing wild-type and Motif 1 mutants of ribosomal protein (RP) gene promoter driving luciferase reporters. **b** Schematic diagram showing RP gene luciferase reporter assay. *RpIII128* promoter lacks Motif 1 and serves as a transfection control. **c** Firefly/Renilla luciferase ratio of relative light unit measurements. Experiments are performed in indicated knockdown conditions. Data represented as mean ± SD from *n* = 3 biological replicates, and *P* values were analyzed by two-tailed unpaired *t*-test. WT *RpLP1* Mock vs *M1BP^RNAi^*; Mut Motif 1 *RpLP1* Mock vs *M1BP^RNAi^*; WT *RpLP30*: Mock vs *M1BP^RNAi^*; WT *RpLP1* Mock vs *Cp190^RNAi^*; and WT *RpLP30* Mock vs *Cp190^RNAi^* all *P* = 0.0001, Mut Motif 1 *RpLP30* Mock vs *M1BP^RNAi^* *P* = 0.0017, Mut Motif 1 *RpLP1* Mock vs *Cp190^RNAi^* *P* = 0.73, Mut Motif 1 *RpLP30* Mock vs *Cp190^RNAi^* *P* = 0.89, WT *RpLP1* Mock vs *mod(mdg4)^RNAi^* *P* = 0.49, Mut Motif 1 *RpLP1* Mock vs *mod(mdg4)^RNAi^* *P* = 0.16, WT *RpLP30* Mock vs *mod(mdg4)^RNAi^* *P* = 0.28, Mut Motif 1 *RpLP30* Mock vs *mod(mdg4)^RNAi^* *P* = 0.98.

with M1BP to activate transcription (Fig. 4b and Supplementary Fig. 4a).

Intriguingly, we found that depletion of CP190 results in similar changes in nascent transcript levels compared to depletion of M1BP. After *Cp190* knockdown, 1894 genes are upregulated, and 1382 genes are downregulated (Fig. 4c). Overall, the extent of nascent transcription changes is correlated with that of *M1BP* knockdown (R = 0.69), with 901 upregulated and 319 down-regulated genes in common (Fig. 4e). In contrast, knockdown of *mod(mdg4)* resulted in little overlap of nascent transcription changes compared to *M1BP* knockdown (R = 0.24, Fig. 4f) or to *Cp190* knockdown (R = 0.29, Fig. 4g). Importantly, CP190 binding at the promoter in the control condition is also enriched at downregulated genes (73%, FET, P < 2.2e–16, odds ratio = 4.1) but not upregulated genes (41%, FET, P = 9.5e–02, odds ratio = 0.9) after CP190 depletion (Fig. 4c). M1BP is also associated with promoters of downregulated genes (49%, FET, P < 2.2e–16, odds ratio = 3.4) but not upregulated genes (21%, FET, P = 4.1e–04, odds ratio = 0.8) after *Cp190* knockdown (Fig. 4d). Finally, we verified that both M1BP plus CP190 binding together are also similarly statistically enriched predominantly at downregulated genes in either *M1BP* (43%, FET, P < 2.2e–16, odds ratio = 2.9) or *Cp190* knockdown condition (44%, FET, P < 2.2e–16, odds ratio = 3.3) (Supplementary Fig. 4), suggesting a direct and specific relationship between CP190 and M1BP in transcriptional activation in particular. Although M1BP and CP190 appear to negatively regulate a similar set of genes, neither M1BP nor CP190 binding is enriched at the promoters of upregulated genes; therefore, these effects are likely to be indirect. Finally, we verified the association of M1BP and CP190 at promoters of several common downregulated genes dependent on M1BP and CP190 using directed ChIP-qPCR (Fig. 4h).

**M1BP and CP190 facilitate Motif 1-dependent gene expression.** To investigate whether CP190 is important for expression of genes dependent on Motif 1, we performed plasmid-based luciferase reporter assays in transfected Kc cells. We monitored luciferase expression driven by Motif 1-containing *RpLP1* or *RpL30* promoters relative to *Renilla*, driven by the Motif 1-independent

*RpIII128* promoter, and expressed on a co-transfected control plasmid (Fig. 5a, b). Cells were mock transfected or knocked down for *M1BP*, *Cp190*, or *mod(mdg4)*. Both M1BP and CP190 depletion decreased Motif 1-dependent *RpLP1* and *RpL30*-driven luciferase expression, while *mod(mdg4)67.2* knockdown had no effect (Fig. 5c and Supplementary Fig. 5). Importantly, *Cp190* knockdown did not affect luciferase expression from constructs driven by *RpLP1* or *RpL30* promoters that harbor point mutations in Motif 1 that abolish M1BP binding[30,31]. These results suggest that CP190 is important for Motif 1-dependent expression and raise the possibility that CP190 may also affect M1BP association with Motif 1-containing promoters.

**CP190 contributes to M1BP chromatin association.** In order to test whether CP190 does indeed play a role in recruitment of M1BP to chromatin, we performed ChIP-seq of M1BP after depletion of CP190 in Kc cells. Western blot analysis illustrated that knockdown of *Cp190* successfully reduced CP190 levels but did not have any effect on M1BP protein levels (Fig. 6a). In addition, we observed no major effect on cell proliferation or indication of mitotic arrest (Supplementary Fig. 2c–e). Using ChIP-seq, we verified substantial reduction of CP190 chromatin association (Fig. 6b). In order to identify statistically significant signal loss of called peaks by ChIP-seq, we applied the DiffBind algorithm[39] at P value <0.05 and identified 806 M1BP sites (22% of total) with significantly decreased M1BP binding after CP190 depletion. This corresponds to loss of M1BP binding at 18% of co-occupied sites. The majority (81%) of reduced M1BP peaks colocalize with CP190 binding in the control condition, suggesting that many effects are direct and that M1BP binding at these sites is facilitated by CP190. We validated six CP190-dependent M1BP sites by directed ChIP-qPCR (Fig. 6c, sites 1–5) and another six sites that are bound by both M1BP and CP190 at which M1BP remains unaffected after CP190 depletion (Fig. 6c, sites 6–11, and Supplementary Fig. 4d). These data suggest that CP190 affects recruitment of M1BP to a subset of its binding sites within the genome.

We next examined the genome-wide relationship between both gene expression and M1BP chromatin association that is

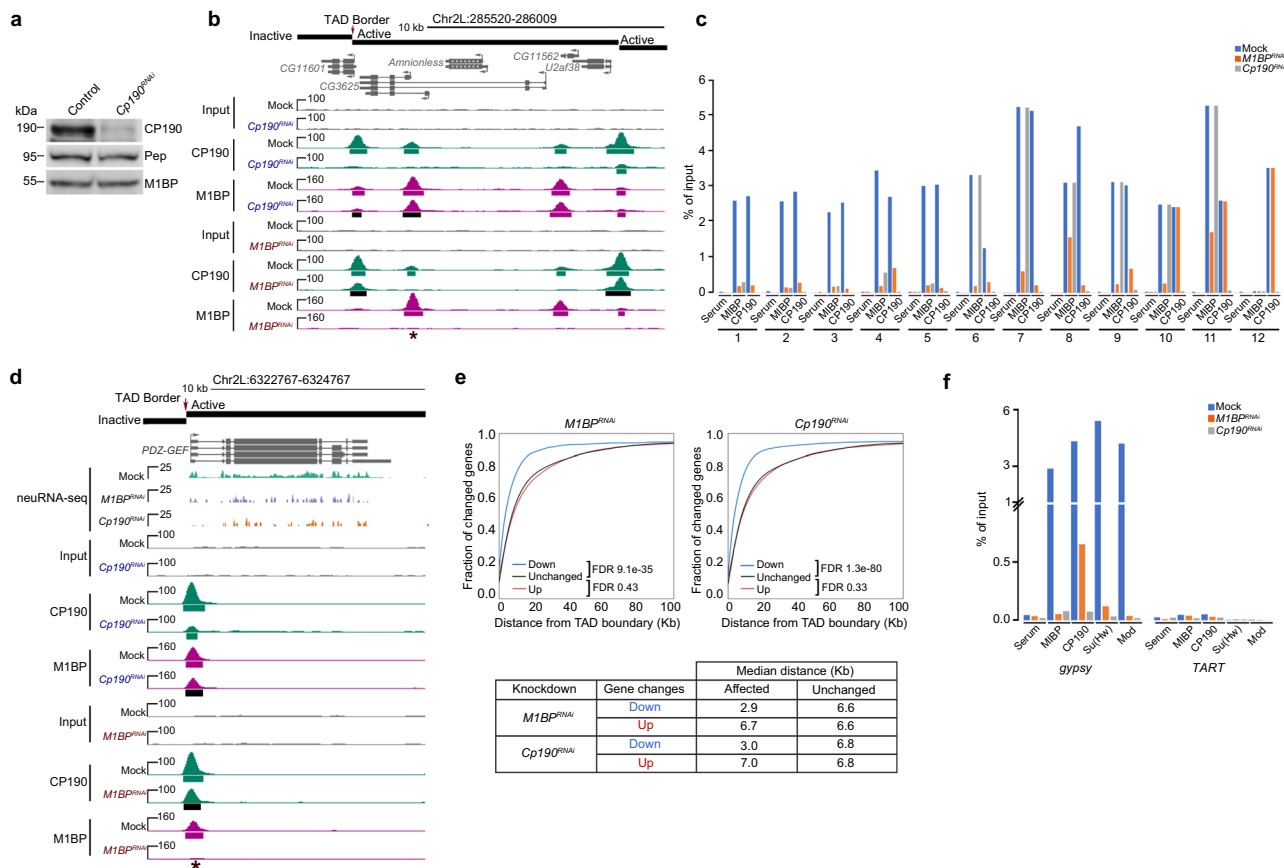

**Fig. 6 M1BP association with chromatin is facilitated by CP190 at a subset of sites. a** Western blotting of total lysates from Kc control and *Cp190* knockdown cells with Pep used as loading control. This experiment was performed for three independent biological replicates with similar results. **b** Example screenshot of ChIP-seq profiles for lost CP190 peak in *M1BP^RNAi* and lost M1BP peak in *Cp190^RNAi* are shown. Asterisk indicates the particular interdependent peak measured in (**c**), site 1. Called ChIP-seq peaks in either knockdown condition shown in black are significantly decreased in the knockdown of the opposite factor. **c** Differentially bound M1BP and CP190 ChIP-seq peaks in *M1BP* knockdown and *Cp190* knockdown validated by ChIP-qPCR. Validation of selected differentially decreased peaks of M1BP and CP190 (sites 1–5) and negative control sites (sites 10 and 11). Validation of decreased peaks of CP190 (sites 6–9) and negative control site 12. Percentage input chromatin DNA precipitated is shown for each primer set, and average values from *n* = 2 biological replicates measured using four technical replicates are plotted. Detailed description of each site labeled is summarized in Supplementary Table 5 and $C_t$ values are available in Source Data 7. **d** Example screenshot showing promoter association of M1BP and CP190 is interdependent for downregulated genes in either knockdown of *M1BP* or *Cp190*. In most cases, both proteins localize at TAD borders. For simplicity, only replicate 1 of mock, *M1BP^RNAi*, and *Cp190^RNAi* is shown for neuRNA-seq tracks. **e** Cumulative histograms of promoter distance from closest TAD border classified by change in nascent expression in *M1BP* (left) or *Cp190* (right) knockdown cells. Downregulated (blue), upregulated (red), or unchanged (black) genes are indicated. Table indicates the median distance from closest TAD border for down- and upregulated genes compared to unchanged genes in either knockdown condition. Mann–Whitney *U* test for each set of changed genes against unchanged genes are shown. Analysis of TADs classified by chromatin state is shown in Supplementary Fig. 6. **f** Percentage of precipitated input chromatin DNA from ChIP-qPCR of M1BP, Su(Hw), Mod(mdg4)67.2, and CP190 at Su(Hw)-binding sites of *gypsy* or *TART* transposon sites as a negative control in Kc cells either mock-treated or subjected to *M1BP^RNAi* or *Cp190^RNAi*. Average of *n* = 2 biological replicates measured using four technical replicates are plotted. $C_t$ values are provided in Source Data 7.

dependent on CP190. We found that CP190-dependent M1BP binding sites are enriched at genes that require CP190 for full expression (Fig. 6d, 9% of genes, FET, *P* = 1.7e–10, odds ratio = 2.0) but not for genes that are upregulated when CP190 is depleted (5% of genes, FET, *P* = 7.0e–01, odds ratio = 0.9). Taken together, CP190-dependent reduction of M1BP at these sites may culminate in a reduction of gene expression.

**CP190 binding at some sites is dependent on M1BP.** We next tested the opposite scenario: the non-exclusive possibility that M1BP affects CP190 recruitment throughout the genome. We thus performed ChIP-seq analysis of M1BP or CP190 in mock-treated versus M1BP-depleted Kc cells. Using the DiffBind

algorithm, we identified 2018 CP190 peaks (25% of total) that were reduced after M1BP depletion (Fig. 3b), and 34% of co-occupied sites show a decrease of CP190 binding. Again, we found that decreased CP190 peaks overlapped with M1BP binding in the control condition in a large proportion of cases (42%, FET, *P* < 2.2e–16, odds ratio = 2.0), suggesting that direct effects are observed. We validated a decrease of both CP190 and M1BP binding at co-occupied sites (sites 1–5) in M1BP knockdown using directed ChIP-qPCR (Fig. 6c and Supplementary Table 5). We also verified one CP190 binding site lacking M1BP that remained unchanged after M1BP depletion (Fig. 6c, site 12, Supplementary Fig. 4c, and Supplementary Table 5).

We then compared the genome-wide relationship between M1BP-dependent CP190 chromatin association and M1BP-

dependent changes in nascent transcription. M1BP-dependent CP190 binding sites are enriched at promoters of genes that require M1BP for full expression (19% of genes, FET, $P =$ 4.3e−09, odds ratio = 1.9) but not genes that are upregulated in *M1BP* knockdown (10% of genes, FET, $P =$ 4.6e−01, odds ratio = 0.9). These results suggest that M1BP directly facilitates CP190 recruitment throughout the genome, and loss of these factors may manifest in reduced gene expression.

**Effects of M1BP depletion occur near TAD borders.** Because both M1BP and CP190 are enriched at promoters of genes located near TAD borders throughout the genome, we wanted to verify whether M1BP and CP190 activate transcription in proximity of TAD borders. We examined the distance from the promoter to the nearest TAD border for either upregulated or downregulated genes in either *M1BP* or *Cp190* knockdown cells and found that genes that require M1BP (false discovery rate [FDR] 9.1e−35) or CP190 (FDR 1.3e−80) for full activation are positioned significantly closer to Kc TAD borders compared to unchanged genes (Fig. 6e). In contrast, genes that are upregulated after *M1BP* (FDR 0.43) or *Cp190* (FDR 0.33) knockdown are located a similar distance from TAD borders compared to unaffected genes. Similar trends are observed for genes located in all four classes of TADs (active, inactive, PcG, and HP1, Supplementary Fig. 6) as previously defined[33]. Overall our observations suggest that M1BP and CP190 promote transcription near TAD borders genome-wide independently of TAD chromatin state.

**M1BP and CP190 promote *gypsy* complex recruitment.** Since M1BP rarely overlaps with Su(Hw) and Mod(mdg4)67.2 across the genome, we wanted to determine the mechanism by which M1BP promotes *gypsy* insulator activity in the context of the retrotransposon. To this end, we examined whether M1BP is also responsible for CP190 recruitment to *gypsy* retrotransposon sites. Interestingly, we did detect substantial M1BP chromatin association along with the three core insulator components at the 12 Su(Hw)-binding sites of the *gypsy* retrotransposon in Kc cells (Fig. 6f). No binding of M1BP or core insulator proteins is observed at *TART*, another element found at multiple locations throughout the genome. After depletion of either M1BP or CP190, chromatin association of all four factors is dramatically reduced at the *gypsy* insulator binding site. These results are consistent with a direct effect of M1BP on *gypsy* chromatin insulator activity and on the recruitment of the core insulator components.

**Depletion of M1BP alters formation of insulator bodies.** Since M1BP promotes core complex recruitment to *gypsy* insulator sites as well as enhancer blocking and barrier activities, we examined the effect of M1BP on the nuclear localization of *gypsy* insulator bodies, of which proper formation correlates with insulator activities. We performed whole-mount immunostaining of dissected brains and imaginal discs of third instar larvae using antibodies against CP190 to detect insulator body localization. In the control line, approximately one insulator body per focal plane was observed in brain optic lobe, eye, leg, and wing discs (Fig. 7a). In contrast, multiple smaller insulator bodies were observed in *M1BP^RNAi* driven by *Act5C-Gal4*, and the differences in number and size of insulator bodies are statistically significant (Fig. 7b, c). Moreover, the combined size of total insulator bodies per nucleus increased significantly in M1BP-depleted compared to control cells (Fig. 7d). These results indicate that M1BP ubiquitously affects localization of insulator bodies.

**M1BP regulates local genome compaction.** Changes observed in insulator body organization prompted us to examine whether M1BP or CP190 may affect global nuclear organization. Therefore, we examined the 3D localization of chromosome territories (CTs) for chromosome arms 2L and 2R using Oligopaint FISH in Kc control versus *M1BP* or *Cp190* knockdown cells. No significant differences in territory volume relative to nuclear volume, CT contact, or CT intermingling were observed (Supplementary Fig. 7). Furthermore, we analyzed the relative spatial positioning of two 50 kb loci (probes A and B) located ~3.1 Mb apart on chromosome 3L and found no difference in the distance between these regions after M1BP- or CP190 depletion (Supplementary Fig. 8). We conclude that M1BP and CP190 do not affect large-scale genome organization.

To investigate if local compaction is altered after M1BP or CP190 depletion, we next measured the distance between two more closely spaced regions (~44 kb apart) in distinct TADs separated by M1BP and CP190 binding sites. These regions are also of interest because M1BP- and CP190-dependent nascent transcription is observed within this vicinity (Supplementary Fig. 9a). Examination of these two 30 kb probes (C and D) in control and M1BP- or CP190-depleted cells showed that depletion of M1BP but not CP190 resulted in a statistically significant decrease in the distance between probes (Supplementary Fig. 9c, e). Moreover, we examined a second pair of probes (E and F) located ~13 kb apart spanning the previously characterized Nhomie and Homie insulator sites respectively[40], which flank a Polycomb Group (PcG)-repressed TAD and also feature M1BP and CP190 binding as well as M1BP- and CP190-dependent nascent transcription (Supplementary Fig. 9b). The distance between these probes was also significantly decreased in M1BP- but not CP190-depleted cells (Supplementary Fig. 9d, e).

Changes in local genome compaction in M1BP-depleted cells could be a result of compaction specifically at TAD borders. To test this possibility, we measured the distances between three 32 kb probes each spaced 15 kb apart with two probes (G and H) inside a large inactive TAD and the third (probe I) spanning the entirety of a flanking PcG TAD (Fig. 8a). We found that distances between G-H and H-I significantly decreased to a similar extent in M1BP- but not CP190-depleted cells, indicating that increased compaction is not exclusively observed across this TAD border but also occurs within this large inactive TAD (Fig. 8b, c). Two additional sets of three 32 kb probes (J, K, L and M, N, O) similarly designed to adjacent TADs likewise show increased inter- and intra-TAD interactions in M1BP- but not CP190-depleted cells (Fig. 8d–i), suggesting that increased local compaction is not limited to TAD borders, specific classes of TADs, or particular configurations.

**Loss of M1BP reduces chromatin accessibility near TAD borders.** Increases in local compaction observed by Oligopaint FISH in M1BP-depleted cells motivated us to examine whether M1BP affects chromatin accessibility genome-wide. We therefore performed Assay for Transposase-Accessible Chromatin (ATAC-seq) in mock-treated or M1BP-depleted Kc cells. We identified 21,355 high-confidence ATAC-seq peaks in the control sample but only 17,585 ATAC-seq peaks in M1BP-depleted cells, indicating substantial overall loss of genome-wide chromatin accessibility and increased chromatin compaction after loss of M1BP. Almost all M1BP ChIP-seq peaks (92%) overlap accessible sites (Fig. 9a), and the average ATAC-seq signal centered on these regions is reduced after M1BP-depletion (Fig. 9b), suggesting that M1BP is required to maintain open chromatin at these sites.

Consistent with M1BP mainly promoting chromatin accessibility, approximately half of M1BP binding sites show a reduction

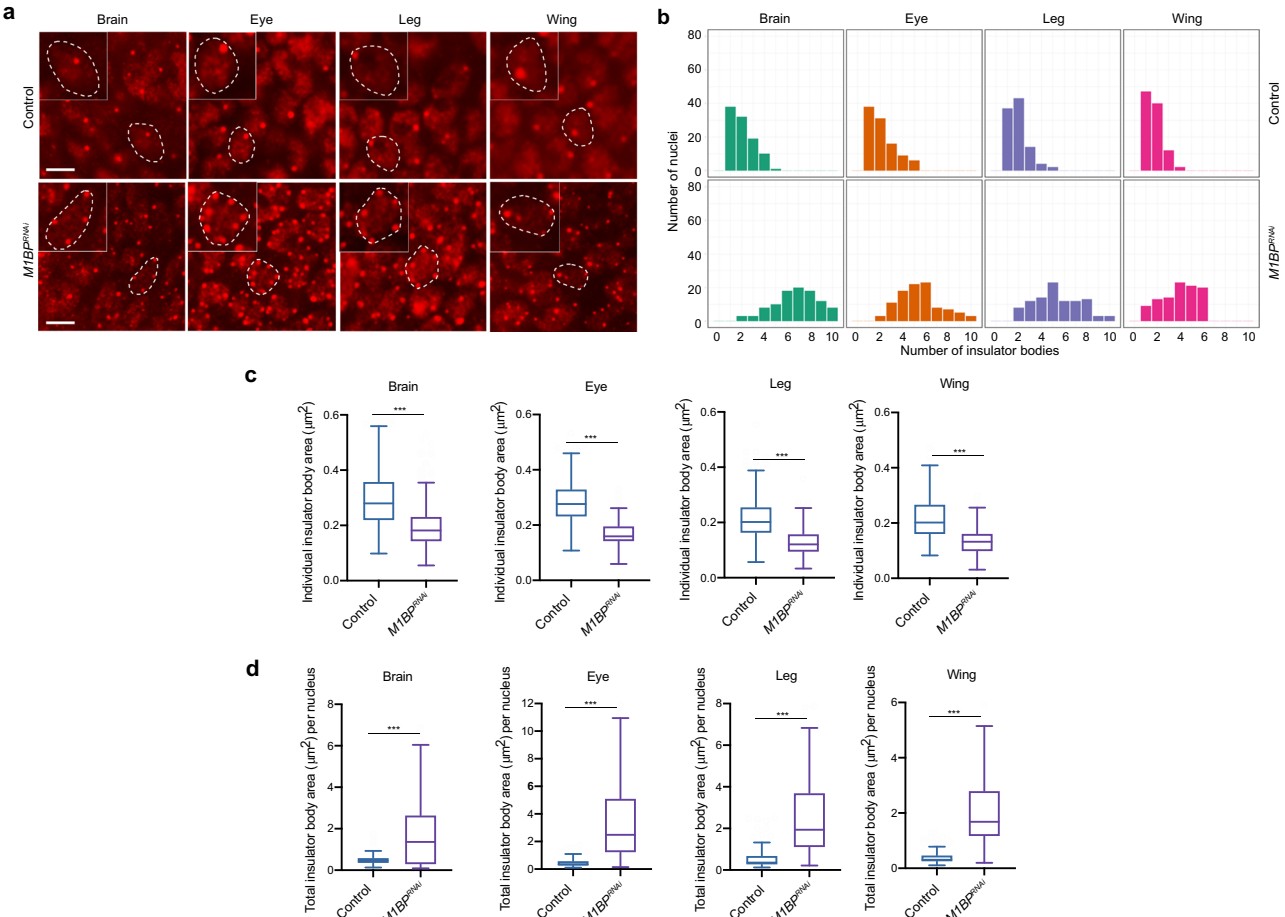

**Fig. 7 Knockdown of *M1BP* alters nuclear organization of insulator bodies. a** Epifluorescence imaging of insulator body localization using anti-CP190 in whole-mount brain, eye, leg, or wing imaginal disc tissues. *M1BP* knockdown is driven by *Act5C-Gal4* driver. Insets show zoom of single nucleus outlined with dashed line in larger panel. Scale bars: 5 μm. Immunostaining experiments were performed three times with similar results. **b** Histograms showing the number of insulator bodies per nucleus in the experiment exemplified in (**a**). In all tissues, the number of insulator bodies is statistically significantly increased in *M1BP* knockdown (Kruskal–Wallis test; all Benjamini–Hochberg corrected $P < 5 \times 10^{-17}$, $n = 100$). **c** Area measurements of individual insulator bodies in brain, eye, leg, and wing imaginal disc tissues of control and *M1BP* knockdown larvae are shown. Bodies were measured (Tukey plots with outliers omitted, Mann–Whitney test $P < 0.001$, $n = 200$). **d** Area measurements of total insulator bodies per nucleus are shown (Tukey plots with outliers omitted, Mann–Whitney test $P < 0.001$); Brain: (Control $n = 108$, *M1BP*<sup>RNAi</sup> $n = 109$), Eye: (Control $n = 121$, *M1BP*<sup>RNAi</sup> $n = 168$), Leg: (Control $n = 107$, *M1BP*<sup>RNAi</sup> $n = 103$), Wing: (Control $n = 109$, *M1BP*<sup>RNAi</sup> $n = 109$). Note that not all cells have discernible nuclear demarcations. **c**, **d** Data are presented as boxplots where box represents the 25–75th percentiles and middle line is the median. The upper whisker extends from the hinge to the largest value no further than 1.5 × IQR from the hinge (where IQR is the interquartile range), and the lower whisker extends from the hinge to the smallest value at most 1.5 × IQR of the hinge, while data beyond the end of the whiskers are outlying points that are omitted from the plots.

of chromatin accessibility as defined by the DiffBind algorithm (FDR < 0.05), whereas approximately 30% correspond to an increase (Fig. 9a). We also observed lower ATAC-seq signal at regions immediately upstream and at well-positioned nucleosomes downstream of the TSS of genes downregulated but not upregulated after depletion of M1BP (Fig. 9c–e), consistent with the previous finding that M1BP activates transcriptionally paused genes[30]. Furthermore, sites with reduced chromatin accessibility overlap significantly specifically with decreased nascent expression (Supplementary Fig. 11a, b). Finally, reduced ATAC-seq signal is observed at TAD borders after M1BP-depletion (Fig. 9f), and reduced ATAC-seq peaks are positioned substantially closer to TAD borders than unchanged or increased peaks (Fig. 9g). In contrast, depletion of CP190 resulted in only mild changes in chromatin accessibility genome-wide (21,093 total ATAC-seq peaks) (Supplementary Fig. 11c–j). Taken together, these results show that loss of M1BP mainly results in reduced chromatin accessibility at its binding sites and in the vicinity of TAD borders

across the genome, and these changes correlate with reduced transcriptional activation.

## Discussion

Here we show that M1BP is required for proper *gypsy* insulator function and insulator body formation and that M1BP and CP190 together activate transcription at TAD borders. We found that M1BP physically associates with CP190 as well as core *gypsy* components and promotes enhancer blocking and barrier activities. Genome-wide M1BP colocalizes mainly with CP190 at Motif 1-containing promoters, which are enriched at TAD borders. M1BP is required for CP190 binding at many sites throughout the genome and vice versa, and loss of either factor reduces gene expression at TAD borders. M1BP is required for proper nuclear localization of insulator bodies, and loss of M1BP increases local genome compaction across TAD borders as well as within large TADs. Finally, M1BP promotes local chromatin accessibility at its

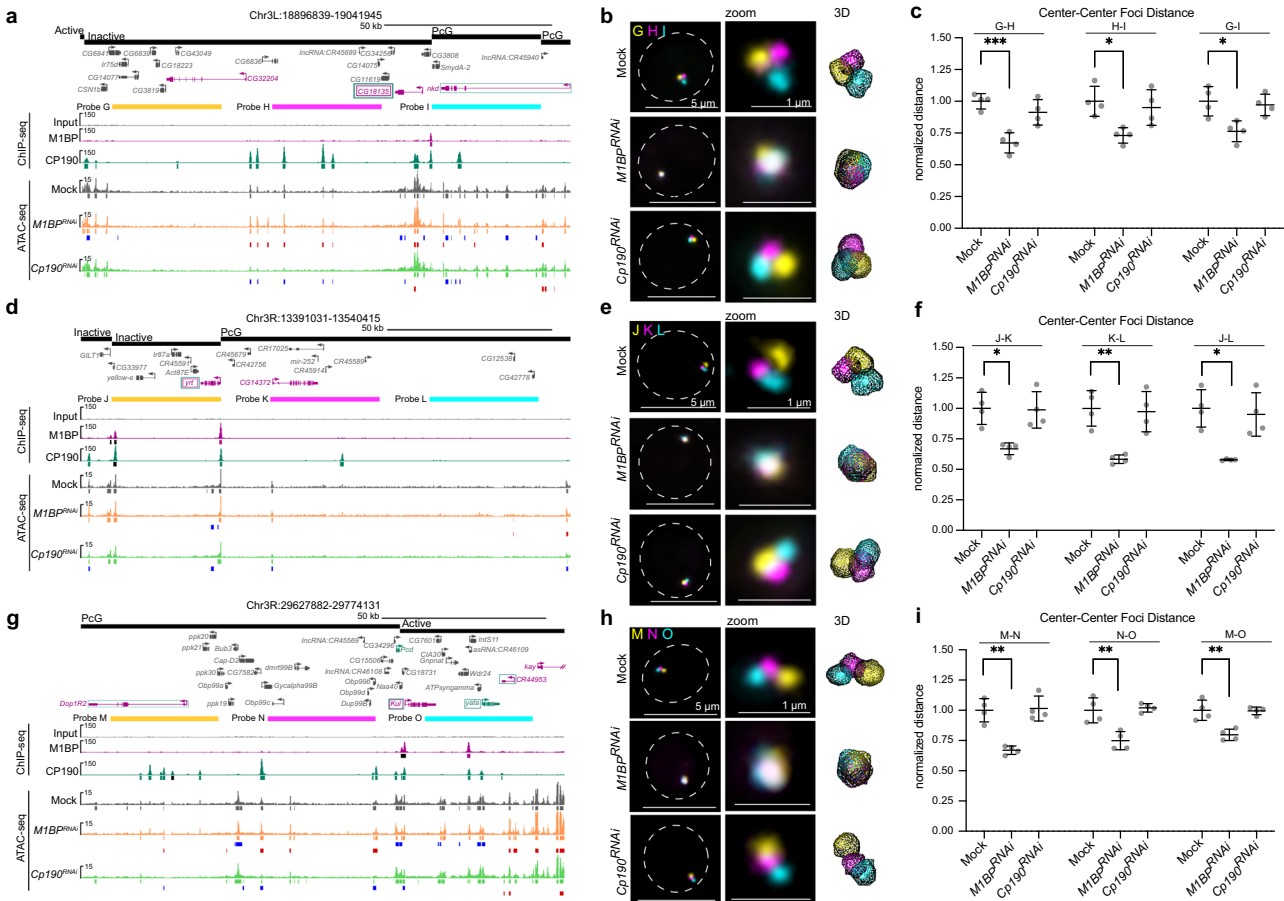

**Fig. 8 Knockdown of *M1BP* increases local inter-TAD and intra-TAD genome compaction. a**, **d**, **g** Regions detected by 32 kb probes spaced 15 kb apart (**a** probes G, H, I; **d** probes J, K, L; **g** probes M, N, O). TADs with state classification, longest gene isoform, ChIP-seq signals, and called peaks (black significantly decreased in knockdown of opposite factor) are shown. ATAC-seq signals and called peaks, decreased peaks relative to mock (blue), and increased peaks (red) are shown. Upregulated genes in either *M1BP* or *Cp190* knockdown are shown in purple or green, respectively. Downregulated genes in either *M1BP* or *Cp190* knockdowns text outlined with a box are shown in purple or green, respectively. Note that *nkd*, *Dop1R2*, and *CR44953* are upregulated in both knockdowns, and *CG18135*, *yrt* and *Kul* are downregulated in both knockdowns. **b**, **e**, **h** Left: representative nuclei labeled with probes in **a**, **d**, and **g**, respectively. Max projection of approximately 5 Z slices. Dashed line represents nuclear edge. Center: zoom of FISH signals. Right: TANGO 3D mesh rendering. **c**, **f**, **i** Dot plots showing average pairwise center-to-center distances between probes in **a**, **d**, and **g**, respectively. Data represented as mean of all replicates (mid-line) ± SD (error bars). Each dot represents the average of one replicate. Single-cell distances were normalized to nuclear radius before population averages were calculated. All averages were normalized to average of mock controls. Only cells in G1 were measured for these analyses. Data are from four biological replicates. See Supplementary Tables 10–12 for "*n*" of cells examined per replicate. Unpaired *t*-test (two-tailed) of means before normalization to controls (*P < 0.05, **P < 0.01, ***P < 0.001). Exact P values can be found in Supplementary Table 13.

binding sites, including transcriptionally activated genes and regions near TAD borders. Taken together, our findings suggest that M1BP may play a role in 3D genome organization through a CP190- and transcription-dependent mechanism.

As M1BP is ubiquitously expressed throughout development, we observe effects on insulator activity and complex localization after M1BP depletion in all tissues and stages of development tested. M1BP associates physically with chromatin at the Su(Hw)-binding sites of the *gypsy* insulator in conjunction with core insulator proteins, and M1BP is required for the binding of all three factors. These findings suggest that M1BP directly affects insulator activity by aiding the recruitment of *gypsy* core components to *gypsy* insulator sites, all three of which are required for proper insulator activity[7,9,10]. Interestingly, Motif 1 is not present at this sequence, and M1BP binding at this site is also dependent on the presence of CP190. One scenario is that binding of the two factors could be cooperative, and M1BP recruitment may additionally help stabilize the multimerization and/or higher order organization of insulator complexes. Consistent with this

hypothesis, depletion of M1BP results in increased numbers of smaller insulator bodies, similar to the effect of complete loss of the BTB-containing core insulator protein Mod(mdg4)67.2[23,26]. Another possibility is that depletion of M1BP results in cellular stress that induces insulator body formation. Since CP190 is a universal insulator protein in *Drosophila*, mislocalization of CP190 may result in, or at least serve as an indicator of, disrupted genome organization when M1BP is depleted.

Although M1BP physically interacts with each of the core *gypsy* insulator components, we observed that M1BP colocalizes mainly with just CP190 throughout the genome, particularly at Motif 1-containing promoters. Distinct binding of the transcriptional activator M1BP compared to Su(Hw) and Mod(mdg4)67.2 genome-wide is not entirely surprising considering sub-stoichiometric levels of co-immunoprecipitation that could also reflect interaction off of chromatin. Furthermore, it has been observed that Su(Hw) binding can correlate with transcriptional repression rather than insulator activity, which may depend on the presence, or absence, of particular interacting proteins[41]. Importantly, recruitment of CP190 is

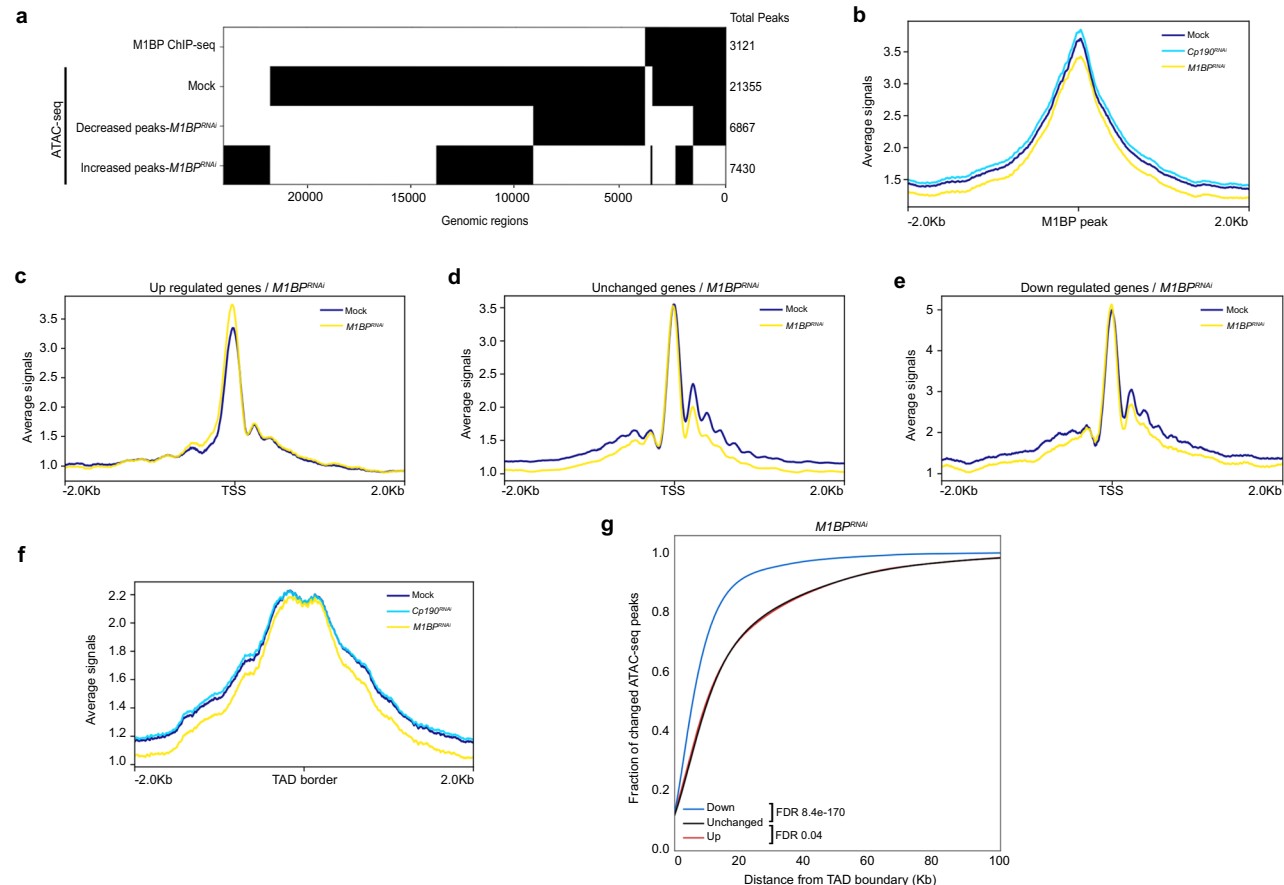

**Fig. 9 Depletion of M1BP reduces the chromatin accessibility near TAD borders genome-wide. a** Binary heatmap of M1BP ChIP-seq peaks, Mock ATAC-seq peaks, decreased and increased ATAC-seq peaks of *M1BP^RNAi* compared to Mock ordered by supervised hierarchical clustering. Each row represents a single independent genomic location, and a black mark in a column represents the presence of a particular factor. **b** Average ATAC-seq tagmentation signals of Mock, *M1BP^RNAi* and *Cp190^RNAi* cells for a 4 Kb. genomic window centered on M1BP-binding sites (ChIP-seq peaks). **c** Average ATAC-seq signals of Mock and *M1BP^RNAi* cells for a 4 Kb genomic window centered on TSS of upregulated genes, **d** unchanged genes, and **e** downregulated genes in *M1BP* knockdown**. f** Average ATAC-seq signals of Mock, *M1BP^RNAi*, and *Cp190^RNAi* cells are plotted for a 4 Kb genomic window centered at TAD borders. **g** Cumulative histograms of ATAC-seq peak center distance from closest TAD border classified by change in ATAC-seq peaks in *M1BP* knockdown cells relative to Mock. Decreased (blue), increased (red), or unchanged (black) ATAC-seq peaks are indicated. Mann–Whitney *U* test for each set of changed peaks against unchanged peaks is shown.

dependent on M1BP and vice versa at many co-occupied sites genome-wide. Why M1BP is only partially dependent on CP190 for binding is unclear, but these results are consistent with the known ability of M1BP to bind DNA directly[30], whereas CP190 is believed to require interaction with a specific DNA-binding protein in order to associate with chromatin[8,17,18,42,43]. Aside from the insulator protein BEAF-32, which binds an AT-rich dual core sequence[44], we did not generally observe a large extent of overlap between M1BP and other DNA-binding insulator proteins that have been shown to be involved in recruiting CP190 to DNA. Future studies may reveal a possible functional relationship between M1BP and BEAF-32 in insulator activity or regulation of gene expression.

Our results suggest that M1BP promotes *gypsy* insulator function through interaction with CP190 in a manner distinct from CP190 interaction with the zinc-finger DNA-binding protein CLAMP. We did not observe a large extent of genome-wide overlap between M1BP and CLAMP, a recently identified positive regulator of *gypsy* insulator activity[16]. The sequence binding specificity of CLAMP is similar to that of GAF[45], while M1BP and GAF bind to and regulate distinct sets of promoters[30]. Although either CLAMP or M1BP depletion reduces *gypsy* enhancer blocking and barrier activities as well as alter insulator body localization, unlike M1BP, CLAMP depletion does not affect

CP190 chromatin association throughout the genome[16]. In fact, CP190 depletion had a substantial effect on CLAMP chromatin association, again suggesting that CP190 may affect the ability of certain DNA-binding proteins, including M1BP, to associate with chromatin through cooperative or higher order physical interactions.

Our results suggest that CP190 may play a more direct role in transcriptional regulation than previously appreciated, in part through interaction with M1BP. Genome-wide profiling studies have shown that CP190 preferentially associates with promoters genome-wide[13,16,46]. CP190 was found to be enriched particularly at active promoters and was shown to affect steady-state gene expression when depleted; however, direct and indirect effects as well as transcriptional and posttranscriptional effects could not be separated[46]. In order to avoid the complication of interpreting steady-state gene expression profiles, we performed neuRNA-seq after either CP190 or M1BP depletion in order to measure newly synthesized transcripts. Intriguingly, nascent RNA expression profiles of CP190 or M1BP-depleted cells showed a remarkably high level of correlation. Since both M1BP and CP190 are particularly associated with promoters of genes that require either factor for adequate expression, it is likely that both factors mainly function in transcriptional activation rather than repression.

M1BP was previously shown to activate transcriptionally paused genes[30], and our ATAC-seq analysis of M1BP-depleted cells supports this conclusion and further demonstrates that M1BP promotes chromatin accessibility surrounding the TSS. Interestingly, depletion of CP190 has no effect on promoter accessibility. Furthermore, we found that CP190 is specifically required for Motif 1-dependent expression of two previously characterized ribosomal protein genes. Whether TRF2 recruitment to Motif 1-containing promoters is affected by CP190 as well as the precise mechanism by which CP190 contributes to M1BP-dependent transcription will be important topics of further study.

We found that genes that require M1BP and CP190 for adequate expression are frequently located at TAD borders, as are both proteins. It has previously been proposed that constitutively active transcription, particularly at/near TAD borders (also referred to as "compartmental domains"), may be a defining or at least key feature of overall genome organization in cells[4,6,35] and throughout development[5]. CP190 was previously shown to be associated particularly with Motif 1-containing promoters[29], and CP190 was observed to be specifically enriched at TAD borders[34,35]. Recently, Motif 1 was also found to be an enriched sequence at TAD borders present in Kc cells[33] and is apparent when TAD borders first appear in embryonic development[5], findings consistent with these previous studies. A previous study of M1BP involvement in genome organization provided limited evidence using Hi-C to suggest that chromosome intermingling may be increased after M1BP depletion[33]. However, the extent and duration of M1BP depletion in their study caused a major disruption of cell cycle and cellular growth, thus obscuring interpretation of those results. We did not observe changes in CT intermingling in our Oligopaint FISH experiments in M1BP-depleted G1 cells, nor did we observe any difference in distance between two distant regions on the same chromosome. Therefore, large-scale changes in genome organization were not observed after M1BP depletion.

We did observe increased local genome compaction after M1BP depletion. This finding led us to test whether TAD borders may be specifically disrupted, perhaps leading to fusion of neighboring TADs. Because our Oligopaint FISH probes are limited to a minimum of 30 kb, we were restricted to intra-TAD analysis of larger TADs, which are typically lower in transcriptional activity (PcG, inactive, null). We found that increased compaction occurs both across TAD borders and within large TADs after M1BP depletion. These effects occur in the vicinity of altered local transcription and reduced chromatin accessibility particularly near TAD borders, suggesting that M1BP-dependent transcriptional changes might alter local chromatin structure that culminates in changes in genome compaction in surrounding regions not restricted to TAD borders. However, the extent of reduced chromatin accessibility observed in M1BP-depleted cells by ATAC-seq is modest relative to the genomic sequence space interrogated by FISH, suggesting that loss of accessibility likely does not directly explain increased local genome compaction. In contrast, CP190 depletion affected transcription to a similar degree, although not identically, yet did not result in changes in local compaction or extensive loss of chromatin accessibility genome-wide. These differences perhaps reflect the multi-functional nature of CP190 as a universal insulator protein contributing to opposing forces, or alternatively, effects on transcription in M1BP-depleted cells may be functionally unrelated to increased genome compaction. Our work shows the requirement of M1BP for accurate CP190 binding throughout the genome as well as *gypsy*-dependent chromatin insulator activity and nuclear localization through interaction with CP190 and other core insulator proteins. Overall, our results provide evidence that M1BP and CP190 at TAD borders and perhaps their ability to activate transcription of constitutively expressed genes located at TAD borders, combined with the capacity of M1BP to promote chromatin accessibility, may play a role in genome organization.

M1BP has been shown to activate transcription of genes at which RNA Pol II is transiently paused in the promoter proximal region, and these promoter regions may themselves possess insulator activity. Intriguingly, a previous study showed that stalled Hox promoters, including Motif 1-containing *Abd-B*, possess intrinsic enhancer blocking insulator activity[47]. However, the second paused promoter identified in this study, *Ubx*, does not harbor Motif 1; thus, M1BP may not necessarily be involved in the enhancer blocking activity of all stalled promoters. Recently, it was shown that M1BP promoter binding can prime the recruitment of the Hox protein Abd-A to the promoter in order to release paused Pol II and activate transcription[48]. We find that M1BP is similarly required for CP190 recruitment at a large number of sites throughout the genome and thus propose that M1BP and CP190 are together required to maintain active gene expression near TAD borders. Active transcription and increased accessibility at these sites may be needed for higher order chromatin organization such as TAD insulation, formation of active compartmental domains, and/or proper local genome structure. Future studies will elucidate the precise mechanisms by which M1BP, CP190, and transcription contribute to higher order chromatin organization.

## Methods

**Drosophila strains**. Fly lines were maintained on standard cornmeal medium at 25 °C. We used lines expressing dsRNA against *su(Hw)* (10724 GD)[24] and *M1BP* (110498 KK) from the Vienna Drosophila RNAi Center. *y,w[1118]; P{attP,y[+],w [3'] (60100 KK)* was used as a control for *M1BP* KK RNAi lines. *Act5C-Gal4*, *Mef2-Gal4* and *l(3)31-1-Gal4* driver lines were obtained from Bloomington Drosophila Stock Center. We scored the *ct*[6] phenotype on the first day after eclosion[16,24]. Number of male flies and percentage of male flies for each score are available in Source Data 7. *UAS-luciferase* constructs were inserted into the *attP3* landing site using phiC31 site-specific integration[49]. Protein extracts from anterior thirds of larvae were used for western blotting. Embryos aged 0–24 h were collected from wild-type Oregon R raised at RT in population cages fed with yeast and molasses. Embryos were dechorionated with 50% bleach, washed and stored at –80 °C[50] to produce nuclear extracts.

**Luciferase insulator barrier activity assay**. Insulator barrier activity by luciferase assay was carried out using Bright-Glo™ Luciferase Assay System (Promega)[16,24]. Individual male larvae were homogenized in 50 μL of Glo Lysis buffer (Promega) and incubated at RT for 10 min. Extracts were cleared from debris by centrifugation. Then, 20 μL of soluble extract was dispensed in a 96-well white flat bottom plate (Costar), and the same volume of Bright-Glo luciferase reagent (Promega) was added to each well. Luciferase signal was quantified using a Spectramax II Gemini EM plate reader (Molecular Devices). Luciferase levels were measured for 12 individual whole third instar male for all genotypes indicated in a single panel simultaneously. Luciferase value was normalized to total protein of each larva determined by BCA reagent (Thermo Scientific). The relative luciferase activity of a population of a single genotype was aggregated into a box and whisker plot. Populations were compared with one-way ANOVA followed by a Tukey HSD post hoc test to obtain *P* values for each pairwise comparison. The *P* values for pairwise comparisons between the control and RNAi lines within both non-insulated and insulated groups are listed in Supplementary Table 4. All values of relative luciferase activity are available in Source Data 7.

**Immunostaining of imaginal discs**. Imaginal discs and brains were dissected from at least six larvae of each genotype following whole-mount staining methods[28]. Dissected tissues were fixed with 4% paraformaldehyde prepared in PBS-T (1X PBS, 0.1% Tween-20) for 20 min with rotation at RT. Tissues were washed 3 × 5 min with PBS-T and blocked with blocking solution (1X PBS, 0.3% Triton, 10% Normal Goat serum) for 2 h. Tissues were immunostained with rabbit serum against CP190 in blocking solution overnight at 4 °C. Tissues were washed 3 × 20 min with PBS-Tr (1X PBS, 0.3% Triton) at RT. Tissues were incubated with secondary antibody in blocking solution at 37 °C for 2 h. Then tissues were washed 2 × 1 h with PBS-Tr, 2 X quickly with PBS-T, incubated with DAPI in PBS-T for 5 min at RT, washed 2 × 5 min with PBS, and mounted with ProLong Diamond mounting media (Life Technologies). Number of insulator foci per nucleus was

counted manually. Number of insulator bodies per nucleus and all area measurements of insulator bodies are available in Source Data 7.

**Co-immunoprecipitation**. Embryonic nuclear extract was prepared from 20 g of mixed stage (0–24 h) *Drosophila* embryos[50]. Frozen embryos were ground in liquid nitrogen with mortar and pestle. The following buffers were prepared, and Roche cOmplete protease inhibitor and 1 mM PMSF were added before use: 4 x NB buffer (240 mM KCl, 60 mM NaCl, 60 mM Tris-HCl pH 7.4, 20 mM MgCl$_2$), NB-1M (1 x NB, 0.1 mM EGTA pH 8.0, 0.5 mM DTT, 1 M sucrose), NB*-1M (1 x NB, 0.5 mM DTT, 1 M sucrose), and NB*-1.7M (1 x NB, 0.5 mM DTT, 1.7 M sucrose). Embryo powder was homogenized with 50 mL NB-1M buffer using glass Kontes tissue grinder and pestle A, and homogenate was cleared by passing through two layered Miracloth at 4 °C. Supernatant was collected by centrifugation at 480 × *g* for 10 min at 4 °C, and the pellet was collected by centrifugation at 4300 × *g* for 10 min at 4 °C. Then, 20 mL of NB*-1.7M buffer was overlaid with 20 mL resuspended pellets in NB*-1M buffer with slight mixing, split into two tubes. Pellets were collected by centrifugation at 19,600 × *g* for 30 min at 4 °C. Nuclei were lysed with 5 mL nuclear lysis buffer HBSMT (50 mM HEPES, pH 6.7; 150 mM NaCl; 5 mM KCl; 2.5 mM MgCl$_2$; 0.3% Triton X-100) supplemented with complete protease inhibitors and 1 mM PMSF and sonicated for 8 cycles with 10 s on and 50 s off[16]. The soluble fraction of extracts was collected by centrifugation. First, two sets of 20 μL of Protein A or Protein G Sepharose beads (GE Healthcare) were washed three times with nuclear lysis buffer for immunoprecipitation (IP) with antibody raised in guinea pig or in rabbit respectively. Then, 3 μL of guinea pig normal serum (Covance Research Products), anti-serum against CP190 (guinea pig), anti-serum against Su(Hw) (guinea pig), rabbit normal serum (Covance Research Products), anti-serum against Mod(mdg4)67.2 (rabbit), or anti-serum against M1BP (rabbit) was incubated with sepharose beads for 1 h at 4 °C, and unbound antibodies were removed by centrifugation at 1500 *g* for 1 min. Beads were washed three times with 0.2 M of sodium borate, pH 9, and then crosslinked with 20 mM DMP in sodium borate for 30 min at RT[51]. Next, beads were collected by centrifugation and washed once with ethanolamine and three times with lysis buffer. After crosslinking, 500 μg of nuclear extract was used for each IP and incubated with antibody-bound beads overnight at 4 °C. The next day, beads were collected by centrifugation and washed two times with nuclear lysis buffer HBSMT and once with HBSM (50 mM HEPES, 150 mM NaCl, 5 mM KCl, 2.5 mM MgCl$_2$). Samples were eluted with SDS sample buffer by boiling, separated using SDS-PAGE, transferred to nitrocellulose membrane in 10 mM CAPS, pH 11, and detected using western blotting. The co-immunoprecipitation efficiency was calculated for each immunoprecipitated protein based on the percentage of total input protein.

**IP and mass spectrometry**. Nuclear extracts from 18 g of mixed stage (0–24 h) *Drosophila* embryos[27] were lysed in 5 mL of HBSMT (50 mM HEPES, pH 6.7; 150 mM NaCl; 5 mM KCl; 2.5 mM MgCl2; 0.3% Triton X-100) including 1 mM PMSF, and Complete protease inhibitor cocktail (Roche). IP was performed using methods described above, and beads were washed twice with HBSMT, once with HBSM, and eluted with 1% sodium dodecanoate. Six replicates for each IP were pooled together for mass spectrometry.

Proteins were analyzed using tandem HPLC-mass spectrometry at the NIDDK Mass Spectrometry Facility. Mass from eluted peptides was queried in the UniProt database, and results were analyzed by MaxQuant v1.6.6.0. Top 50 proteins for each pulldown are shown in Supplementary Tables 1 and 2.

**Cell lines**. Kc167 cells were grown in CCM3 media (Thermo Scientific HyClone, Logan, UT). Cells were maintained in monolayer at 25 °C.

**DsRNA knockdowns**. A total of 1 × 10⁷ Kc167 cells were transfected with either 20 μg of dsRNA against *M1BP*[30], 12 μg of dsRNA against *Cp190*[16,28], 5 μg of dsRNA against *mod(mdg4)* (specific to deplete the Mod(mdg4)67.2 isoform), or no RNA (mock) using Amaxa cell line Nucleofector kit V (Lonza) and electroporated using G-30 program. Cells were incubated for 5 days at 25 °C to obtain efficient depletion of each protein but minimal effects on cell viability and growth. The primers for generating the templates for in vitro synthesis of RNAi are listed in Supplementary Table 6.

**Quantification of cell viability**. Cell viability was quantified by Trypan Blue exclusion test. Then, 10 μL of cells mixed with equal amount of trypan blue was examined to determine the percentage of cells that have clear cytoplasm (viable cells) versus cells that have blue cytoplasm (nonviable cells). Number of viable cells of mock, *M1BP* dsRNA, or *Cp190* dsRNA treated for each day of knockdown were recorded. Viability graphs were generated from the values of viable cell numbers.

**Quantification of cell proliferation and mitotic index**. Cells were collected each day of knockdown for mock, *M1BP* dsRNA, or *Cp190* dsRNA treatments. Cells were immunostained with rabbit antibody against Histone H3 (phospho S10) (Abcam ab5176; 1:5000), mouse anti-Tubulin (Sigma T6074; 1:500), labeled with DAPI stain, and mounted with ProLong Diamond mounting media (Life

Technologies) to assay cell viability and mitotic index. Numbers of mitotic cells were counted manually. Statistical tests were performed using Prism 8 software v8.4.2 by GraphPad.

**FISH with Oligopaints**. Oligopaints were designed to have 80 bp of homology (probes A–F) or 60 bp of homolog (probes G–O) and an average probe density of 5 and 6.5 probes per kb, respectively, using a modified version of the OligoMiner pipeline[52]. Oligopaints were generated in the lab using the T7 method[53]. Briefly, oligo pools were ordered from Twist Bioscience and were amplified by PCR according to instructions from the manufacturer using primers containing the F and R barcode sequences added to oligos during the design process. The T7 sequence was added to the reverse primer (TAATACGACTCACTATAGGG), and secondary oligo binding sites were added to the forward primer. T7-labeled PCR products were then in vitro transcribed using the Hi-Scribe T7 RNA synthesis kit (NEB). RNA was converted to RNA:DNA duplexes by reverse transcription using the Thermo Scientific™ Maxima™ H Minus Reverse Transcriptase. RNA was degraded to leave single-stranded DNA FISH probes using alkaline hydrolysis, and probes were purified using the Zymo-100 DNA Clean and Concentrator Kit. Coordinates for all probes and list of primers can be found in Supplementary Tables 8 and 9, respectively. Chromosome 2L and 2R Oligopaints were a gift from E. Joyce[54].

For FISH, slides were prepared by dispensing 1.5 × 10⁵ cells onto 0.01% poly-L-lysine coated slides and fixed with 4% paraformaldehyde for 15 min at RT, followed by washes two times, each for 5 min in PBS-T. Slides were then permeabilized with PBS-T$^{0.5}$ (0.5% Triton X) for 15 min before storing in 70% ethanol overnight at −20 °C. The next day, slides were dehydrated in an ethanol row (90% ethanol for 5 min at −20 °C, 100% ethanol for 5 min at −20 °C) before allowing them to dry at RT for 5 min. Next, slides were washed once in 2x SSCT for 5 min at RT, once in 2x SSCT/ 50% formamide for 5 min at RT, once in 2x SSCT/ 50% formamide at 92 °C for 2.5 min, and once in 2x SSCT/50% formamide at 60 °C for 20 min. For hybridization, 50–150 pmol of each probe was used per slide in a final volume of 25 μL. After the applying of primary Oligopaint probes, slides were covered with a coverslip and sealed. Denaturation was performed at 92 °C for 2.5 min. Slides were transferred to a 37 °C humidified chamber and incubated for 16–18 h. Then slides were washed in 2x SSCT at 60 °C for 15 min, 2x SSCT at RT for 15 min, and 0.2x SSC at RT for 5 min. Fluorophore conjugated secondary probes (0.4 pmol/μL) were added to slides, covered, and sealed. Slides were incubated at 37 °C for 2 h in a humidified chamber followed by washes as for the primary probes. All slides were washed with DAPI DNA stain for 5 min, followed by 2 × 5 min washes in PBS before mounting in Prolong Diamond (Life Technologies).

**Imaging, quantification, and data analysis**. Images were captured at RT on a Leica DMi 6000B widefield fluorescence microscope using a 1.4 NA 63x or 100x oil-immersion objective and Leica DFC9000 sCMOS Monochrome Camera. DAPI, CY3, CY5, and FITC filter cubes were used for image acquisition. Images were acquired using LasX Premium software (Leica Application Suite X 3.6.0.20104) and deconvolved using Huygens Professional software v19.10 (Scientific Volume Imaging). After deconvolution, images were segmented and measured using the TANGO 3D-segmentation plugin (v 0.99) for ImageJ (v2.0.0-rc-69/1.53cas)[55]. All nuclei were segmented using the "Hysteresis" algorithm within TANGO. For CT measurements, CTs were segmented using the "Spot Detector 3D" algorithm within TANGO. Foci were segmented using the "Hysteresis" algorithm. To avoid confounding effects of possible differences based on cell cycle stage, imaged nuclei were sorted into G1 and G2 cell cycle stages based on nuclear volume as measured in TANGO (where G1 nuclei were less than 150 μm³ and G2 nuclei were between 150 and 300 μm³)[54]. Only G1 nuclei, which would have recently divided during the time of the experiment, were included for all FISH analyses. Statistical tests were performed using Prism 8 (v8.4.2) software by GraphPad. Foci volume measurements are shown in Supplementary Figs 8d and 10. All values from TANGO measurements are available in Source Data 8.

**ChIP and ChIP-seq library preparation**. A total of 2–3 × 10⁷ cells were fixed by adding 1% formaldehyde directly to cells in culture medium for 10 min at RT with gentle agitation. Formaldehyde was quenched with 0.125 M glycine with gentle agitation for 5 min. Then cells were pelleted by centrifugation at 2000 × *g* and washed twice with ice-cold PBS. The pellet was lysed with 0.8 mL ice-cold cell lysis buffer (5 mM PIPES pH 8, 85 mM KCl, 0.5% NP40, Roche cOmplete protease inhibitor). Nuclear pellet was collected by centrifugation at 2000 × *g* for 5 min at 4 °C and resuspended in 1 mL of nuclear lysis buffer (50 mM Tris-HCl pH 8, 10 mM EDTA, 1% SDS, Roche cOmplete protease inhibitor). Then nuclear lysate was incubated for 10 min with rotation at 4 °C. After adding 0.5 mL of IP dilution buffer (16.7 mM Tris-HCl pH 8, 1.2 mM EDTA, 167 mM NaCl, 1.1% Triton X-100, 0.01% SDS, Roche cOmplete protease inhibitor) and 300 mg of acid-washed 212–300 μm glass beads (Sigma-Aldrich) to nuclear lysate, chromatin was fragmented using a Bioruptor (Diagenode) using 10 cycles of 30 s on and 30 s off, maximum output. Lysate was centrifuged at max speed for 10 min at 4 °C, and supernatant (chromatin) was saved at −80 °C. Chromatin was diluted to 1:5 with IP dilution buffer and added to 50 μL prewashed Protein A Sepharose beads (GE Healthcare) and incubated with 5 μL of respective antibody overnight at 4 °C

with rotation. The next day, beads were washed with the following wash buffers: 3× with low-salt wash buffer (20 mM Tris-HCl pH 8, 2 mM EDTA, 150 mM NaCl, 1% Triton X-100, 0.1% SDS), 3× with high-salt wash buffer (20 mM Tris-HCl pH 8, 2 mM EDTA, 500 mM NaCl, 1% Triton X-100, 0.1% SDS), and 2× with LiCl wash buffer (10 mM Tris-HCl pH 8, 1 mM EDTA, 250 mM LiCl, 1% NP40, 1% deoxycholate). Then, ChIP sample was eluted 2× with 200 μL of elution buffer (0.1 M NaHCO3, 1% SDS) for 30 min at 65 °C in a thermomixer at 800 rpm. Eluates were de-crosslinked with solution (20 μL of 5 M NaCl, 8 μL of 0.5 M EDTA, 10 μL of 1 M Tris-HCl pH 8) overnight at 65 °C. The next day, 4 μL of proteinase K (20 mg/mL) was added and incubated for 2 h at 50 °C, purified using phenol-chloroform followed by ethanol precipitation with 2 μL of Glycoblue (Ambion), 0.1 vol of 3 M NaOAc pH 5.2, 2.5 vol of ice-cold 100% ethanol and incubated overnight at −80 °C. Samples were pelleted by centrifugation at high speed 20 min at 4 °C. After washing with 70% ethanol, pellets were air-dried at RT and resuspend in 10 μL of nuclease-free water. Details of antibodies can be found in Supplementary Table 7. Libraries were constructed by pooling two IP samples using TruSeq adapters (Illumina) according to the TruSeq Illumina ChIP-seq sample preparation protocol with minor modification: after adaptor ligation and PCR amplification, samples were purified using AMPure XP Bead (sample: beads ratio 1:0.8) according to the manufacturer's protocol. All samples were sequenced with HiSeq2500 (Illumina) using 50 bp single-end sequencing.

**ChIP-quantitative PCR**. We performed quantitative PCR using ChIP DNA samples of normal serum IP (negative control), M1BP IP, CP190 IP, Su(Hw) IP, and Mod(mdg4)67.2 IP from both mock-treated Kc cells and Kc cells transfected with dsRNA against *M1BP* or *Cp190*. ChIP DNA samples were amplified on an Applied Biosystems real-time PCR machine using site-specific primer sets (Supplementary Table 5) and quantified using SYBR Green (Applied Biosystems) incorporation. Experiments were performed with two independent biological replicates, and each sample was quantified using four technical replicates. $C_t$ values are available in Source Data 7.

**ATAC-seq library preparation**. ATAC-seq was performed following a protocol from the Kaestner Lab (https://www.med.upenn.edu/kaestnerlab/assets/user-content/documents/ATAC-seq%20Protocol%20(Omni)%20-%20Kaestner%20Lab.pdf) with minor modifications. A total of 100,000 Kc cells were washed with 50 μL cold 1X PBS. Cell pellet was lysed with 50 μL cold lysis buffer (10 mM Tris-HCl, pH 7.5, 10 mM NaCl, 3 mM MgCl2, 0.1% NP40, 0.1% Tween-20, 0.01% Digitonin) and incubated for 10 min on ice. Then 500 μL of wash buffer (10 mM Tris-HCl, pH 7.5, 10 mM NaCl, 3 mM MgCl2, 0.1% Tween-20) was added to lysate. Nuclei were then collected by centrifuging at 500 × g for 10 min at 4 °C and nuclei were resuspended in 50 μL of transposition reaction mix [25 μL 2X TD buffer (Illumina), 16.5 μL PBS, 0.5 μL 10% Tween-20, 0.5 μL, 0.5 μL 1% Digitonin, 2.5 uL Tn5 enzyme (Illumina), 5 μL nuclease-free water] and incubated for 45 min at 37 °C at 1000 rpm (Eppendorf Thermomixer) for fragmentation. DNA was purified with Qiagen Minelute columns, and libraries were amplified by adding 10 μL DNA to 25 μL of NEBNext HiFi 2x PCR mix (New England Biolabs) and 2.5 μL of 25 μM each of Ad1 and Ad2 primers using 11 PCR cycles. Libraries were purified with 1.2× AMPure XP beads. All samples were sequenced with NextSeq-550 (Illumina) using 50 bp paired-end sequencing.

**Nascent EU-RNA labeling and library preparation**. Nascent EU RNA-seq (neuRNA-seq) labeling and capture were done by using Click-iT Nascent RNA Capture Kit (Thermo Fisher Scientific) according to the manufacturer's protocol. Kc cells were incubated with 0.2 mM EU for 1 h and RNA was extracted with Trizol (Thermo Fisher Scientific). Next, RNA was chemically fragmented for 5 min at 70 °C with RNA Fragmentation Reagents (Thermo Fisher Scientific), followed by DNase I treatment (Roche). Then RNA was ethanol precipitated after Phenol: Chloroform (Thermo Fisher Scientific) purification. The Click-iT reaction was performed with 0.5 mM biotin azide using 5 μg of EU-RNA, and biotinylated RNA was captured with 12 μL T1 beads. The nascent EU-RNA was used to generate RNA-seq libraries with Ovation RNA-seq Systems 1–16 for Model Organisms (Nugen). Samples were sequenced on HiSeq2500 (Illumina) using 50 bp single-end sequencing at the NIDDK Genomics Core Facility.

**Luciferase reporter assays**. A total of $5 \times 10^6$ Kc cells were transfected with dsRNA against *M1BP*, *Cp190*, and *mod(mdg4)* as described above. After 2 days of incubation with dsRNA at 25 °C, luciferase reporter construct expressing either wild-type or Motif 1 mutant of pGL3-(RpLP1 [−500 to + 50]) or pGL3-(RpLP30 [−500 to + 50]) was co-transfected with RpL-polIII-Renilla into cells. Cells were incubated for an additional 3 days and then assayed in tandem for firefly and Renilla luciferase activity. Cells were lysed and assayed using the dual-luciferase reporter assay system (Promega) according to the manufacturer's protocol. Luciferase activity values are available in Source Data 7.

**ChIP-seq data analysis**. FASTQ files of sequenced single-end 50 bp reads were trimmed using cutadapt v1.8.1[56] with arguments "--quality-cutoff 20," "-a AGATCGGAAGAGC," "--minimum-length 25," and "--overlap 10." Then trimmed reads were mapped to the Flybase r6-24 dm6 genome assembly with Bowtie2

v2.3.5[57] with default arguments. Multimapping reads were removed mapped reads using samtools v1.9[58] view command with the argument -q 20. Duplicates were removed from mapped, uniquely mapping reads with picard MarkDuplicates v2.20.2 (http://broadinstitute.github.io/picard/index.html). MACS2 v2.2.5[59] (https://github.com/taoliu/MACS) was used to call peaks by providing replicate IPs and inputs as multiple BAMs, effectively calling peaks on pooled/merged samples and using additional arguments "-f BAM," "--gsize=dm," "--mfold 3 100" (the latter to include a larger set of preliminary peaks for fragment size estimation).

Binary heatmaps were generated using pybedtools v0.8.1[60,61]. Since peaks for one protein can potentially overlap multiple peaks for other proteins, the output represents unique genomic regions as determined by bedtools multiinter with the -cluster argument. Therefore, as a result, when considering the multi-way overlap with other proteins, the sum of unique genomic regions for a protein is not guaranteed to sum to the total number of called peaks for that protein. Source of called peaks for all other factors are summarized in Supplementary Table 14. Dm3 assembly is lifted over and converted to dm6 (liftover, https://genome.ucsc.edu/cgi-bin/hgLiftOver).

Pairwise comparison for co-localization of different factors was performed using the BEDTOOLS v2.27.1 "jaccard" command[60,61]. The heatmap was clustered with the scipy.cluster.hierarchy module, using "euclidean" as the distance metric and "Ward" as the clustering method. As the Jaccard statistic is independent of the order of comparison and is symmetric across the diagonal, only the upper triangle is shown.

FlyBase release 6.24 annotations were used to annotate peaks falling into genomic regions in Fig. 3d as follows. Exons were defined as any exon from any transcript of any gene. Introns were defined as the space between exons derived in a per-transcript manner by using the gffutils v0.10.1 (https://github.com/daler/gffutils) method FeatureDB.create_introns(). Promoters were defined as the TSS of each transcript plus 1500 bp upstream. Intergenic regions were defined as all regions between gene bodies. Shared co-bound peaks were determined by using pybedtools BedTool.intersect with the v=True or u=True argument, respectively. Each set of peaks was intersected with the annotations using this hierarchy "promoter>exon>intron>intergenic" where a peak was classified according to the highest priority feature. Here, a peak simultaneously intersecting a promoter of one isoform and an intron of a different isoform would be classified as "promoter." To compare percentages across annotated peaks in different types of peaks (All M1BP, all CP190, shared M1BP and CP190 peaks and all *gypsy* sites), the number of peaks in each class was divided by the total number of peaks of that type.

Heatmaps were generated by using default deepTools v3.4.1 packages in Galaxy. ChIP reads were normalized to reads of input samples and mapped in a 3 kb window centered on the TSS containing Motif 1 consensus sequence.

**Differential ChIP-seq**. To detect differential ChIP-seq binding, we used the Diffbind v2.14.0 /R package[39] using the config object "data.frame(RunParallel=-TRUE, DataType=DBA_DATA_FRAME, AnalysisMethod=DBA_EDGER, bCorPlot=FALSE, bUsePval=FALSE, fragmentSize=300)" and otherwise used defaults. Input files consisted of the final peak calls mentioned above and the IP and input BAM files for each replicate as described above with multimappers and duplicates removed. Then the final results were exported with the dba.report function with parameters "th=1, bCalled=TRUE, bNormalized=TRUE, bCounts=TRUE" and final differentially gained or lost peaks were those that had a log2fold change of >0 or <0, respectively, and *P* value <0.05 (R version 4.0.2).

**ATAC-seq data analysis**. The ATAC-seq data were processed based on the ATAC-seq Guidelines (https://informatics.fas.harvard.edu/atac-seq-guidelines.html). The raw paired-end fastq files trimmed by 51 bp reads were trimmed using cutadapt v1.8.1[56] with arguments "--quality-cutoff 20," "-a AATGATACGGCGA CCACCGAGATCTACACTCGTCGGCAGCGTCAGATGTG-AAATGATACGG CGACCACCGAGATCTACACTCGTCGGCAGCGTCAGATGTG," "--minimum-length 18." Then trimmed reads were mapped to the Flybase r6-24 dm6 genome assembly with Bowtie2 v2.3.5[57] with "--very-sensitive -X 2000." chrM reads were removed using samtools v1.9[58]. Duplicates were removed with picard MarkDuplicates v2.20.2 (http://broadinstitute.github.io/picard/index.html). We filtered the unmapped reads, non-primary alignment and kept the proper pair reads and unique mapped reads (-f 2 -q 30) using samtools v1.9. MACS2 v2.2.5[59] (https://github.com/taoliu/MACS) was used to call peaks for only properly paired alignments with "-f BAMPE," "--gsize=dm," "-q 0.0001." For mock, CP190 knockdown and M1BP knockdown, we merged the peak files from three replicates in each condition.

To detect differential ATAC-seq signals, we first merged the peaks in two conditions (CP190 knockdown vs mock and M1BP knockdown vs mock). For ATAC-seq, we did not set "bamControl" in DiffBind. We used the Diffbind v2.14.0/R package[39] using the config object "data.frame(RunParallel=TRUE, DataType=DBA_DATA_FRAME, AnalysisMethod=DBA_EDGER, bCorPlot= FALSE, bUsePval=FALSE)." Input files consisted of the merged peak files and the BAM files for each replicate. Final differentially gained or lost peaks were those that had a log2fold change of >0 or <0, respectively, and FDR < 0.05.

For ATAC-seq peak overlap, we used same method as ChIP-seq to produce binary heatmaps using pybedtools v0.8.1[60,61]. We compared mock ChIP-seq peaks and ATAC-seq peaks.

We applied "computeMatrix" and "plotProfile" in the deeptools package v3.4.1 to plot the ATAC-seq signals around ChIP-seq M1BP or CP190 peaks, TAD borders[33], and TSS of differential genes after *M1BP* or *Cp190* knockdown.

FET was performed using number of gene promoters occupied/not occupied by decreased/increased ATAC-seq peaks from the group of upregulated (log2fold > 0, $P_{adj} < 0.05$), downregulated (log2fold < 0, $P_{adj} < 0.05$), and not affected gene promoters, respectively, ascertained using DESeq2.

**neuRNA-seq (mapping and read counting).** Low-quality bases and adaptors were trimmed from sequence reads using cutadapt v2.7[56] with parameters "–q 20 -minimum-length 25 -a AGATCGGAAGAGCACACGTCTGAACTCCAGTCA." Resulting reads were mapped to the FlyBase r6-24 reference genome using HISAT2 v2.1.0[62] with default parameters. Aligned reads were counted using subread fea-tureCounts v1.6.4[63] with default parameters, except that "-t gene" option was used to quantify the reads on gene feature for EuSeq, and "-s1" option specifically for this sense-stranded library. Differential expression analysis was done with DESeq2 v1.22.1[64] in v3.5.1R version.

**neuRNA-seq (differential expression).** Counts tables were loaded into DESeq2 v1.22.1 for[64] neuRNA-seq analysis. Counts tables from independent neuRNA-seq experiments were independently imported and normalized using simple design "~treatment" and using otherwise default parameters. Differentially expressed genes were those with $P_{adj} < 0.05$.

**Fisher's exact tests.** These tests were performed using number of gene promoters occupied/not occupied by M1BP or CP190 or both from the group of upregulated (log2fold > 0, $P_{adj} < 0.05$), downregulated (log2fold < 0, $P_{adj} < 0.05$) and not affected gene promoters, respectively analyzed from the DESeq2. Prior to determining the number of peaks intersecting with gene promoter regions from the selected group, the promoter regions were merged if there were any overlapping promoter regions. For intersections with ChIP peaks, a gene promoter region was defined as 1 kb up- and downstream of TSS. The Flybase r6.24 gene annotation gtf file was used to define gene start sites.

FET was used to test whether the up/downregulated genes show preference on the overlapped peaks. FET is implemented by Python scipy package v1.5.2 (scipy. stats.fisher_exact). FET and odds ratios are based on the 2 × 2 contingency table. All genes are used. FET test uses two-tailed test.

| Genes | Up/Down | Others | Row total |
|---|---|---|---|
| Overlap with ChIP peaks | a | b | a + b |
| No overlap | c | d | c + d |
| Column total | a + c | b + d | a + b + c + d = n |

The *P* value is computed as follows:

$$P = \frac{(a+b)!(c+d)!(a+c)!(b+d)!}{a!b!c!d!n!} \tag{1}$$

$$\text{The odds ratio} = \frac{a/b}{c/d} \tag{2}$$

**Feature distance analysis to closest TAD border.** Four types of TADs (Active, inactive, HP1, and PcG TADs) were downloaded from http://chorogenome.ie-freiburg.mpg.de/[33] and lifted over to the dm6 assembly (liftover, https://genome.ucsc.edu/cgi-bin/hgLiftOver). The distance of the feature (promoter or ATAC-seq peak) to the TAD border was computed for each type of TAD, and then pybedtools packages including sort and closest operations were used to search the nearest TAD for each feature. Next, features inside the TAD were selected and computed for the minimum distance to the TAD border. Seaborn/0.9.0 package (seaborn.kdeplot) was used to calculate the kernel density of distance for up, down, and unchanged features, and cumulative distribution plots were generated based on the kernel density estimation. The matplotlib/3.1.1 package was used to plot the density of gene distance to TAD border as a histogram (matplotlib.pyplot.hist, bin=100).

**Mann–Whitney *U* test.** The Mann–Whitney *U* test ("two-sided", scipy.stats. mannwhitneyu, scipy v1.5.2) was used for each set of changed genes against unchanged genes to test whether the difference of both mean TAD border distances is significant. FDR correction was performed using multiple test correction (Benjamini–Hochberg method, statsmodels.stats.multitest.multipletests, statsmo-dels v0.12.0).

**Motif 1 derivation.** Motif 1 analysis was derived from the MEME v3.0 analysis of promoters[38]. Motif scanning was carried out using FIMO v5.0.1[65] for all gene promoter regions. The Motif 1_input was a position probability matrix of Motif 1. The position probability matrix of Motif 1 G[A/C]TACGGTCACACTG was obtained by transformation of a position weight matrix from a previous study[38].

The transformation converts the weight matrix to the probability position matrix by using the definition

$$\ln\left(\frac{N_{ij} + p_i}{N+1}\right) - \ln(p_i) = M_{ij}. \tag{3}$$

Here, $M_{ij}$ is the entry of weight matrix and $p_i$ the background probability of nucleotide *i*, and *N* is the total number of sequences used in motif prediction. Assume $p_i$ is equal to 0.25. Since *N* is much larger than $p_i$, the equation can be rewritten into

$$\ln\left(\frac{N_{ij}}{N+1}\right) - \ln(p_i) = M_{ij} \text{ or } \ln(P_{ij}) - \ln(p_i) = M_{ij}. \tag{4}$$

$P_{ij}$ is an entry of position probability matrix[66]. After rounding all entries to the 19th decimal, only the probability of the second position was assigned as 0.5 for nucleotide A and C, respectively. The remaining positions in the matrix were assigned to 1 for each corresponding nucleotide. Other than that, all entries were assigned to zero. The motif search cutoff used was $P \le 7.12e-5$ and shown in Supplementary Data.

**Statistics and reproducibility.** All quantifications were conducted unblinded and no statistical method was used to predetermine sample size. Statistical parameters including exact value of "*n*," "*P*" values, and the types of the statistical tests are reported in the figure legends, and "*n*" of cells examined for four biological replicates for Fig. 8 are summarized in Supplementary Tables 10–12. Error bars are derived from at least three independent biological replicates and displayed on graphs showing either the mean ± SEM or mean ± SD, as indicated in the figure legends. Immunostaining experiments were performed three times with similar results. All box and whisker plot parameters are defined in the respective figure legends. ChIP-qPCR was conducted using two biological replicates each measured using four technical replicates. Statistical analysis was carried out using Prism 8 (GraphPad Software). All statistical tests used for sequencing data analysis are explained in detail in the "Methods" section. All western blotting experiments were performed two to three times with similar results. Co-immunoprecipitation experiments were performed two times with similar results. Quantitative mass-spec analysis was performed by pooling together six replicates for each IP.

**Reporting summary.** Further information on experimental design is available in the Nature Research Reporting Summary linked to this paper.

## Data availability

The accession numbers for the raw data FASTQ files, processed files, and BigWig files for all sequencing data generated in this study and deposited in NCBI GEO are GSE142533 and GSE169105. All other publicly available sequencing data analyzed in this study include Kc167 CLAMP GSM2775116, Kc167 BEAF-32 GSM762845, S2 ZIPIC GSM1313421, S2 Pita GSM1313420, S2 Ibf1 GSM1133264, S2 Ibf2 GSM1133265, and Kc167 CTCF GSM1535983. Mass spectrometry data files have been deposited in the PRIDE repository with dataset identifiers PXD026493 for CP190 and PXD026497 for Su (Hw). All other relevant data of this study are available from the corresponding author upon reasonable request. Source Data are provided with this paper.

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

## Acknowledgements

We thank D. Gilmour for sharing previously unpublished results, rabbit anti-M1BP sera, and for comments on the manuscript. We also thank R. Dale for initial software development, A. Beyer for anti-Pep, P. O'Farrell for anti-Pc antibodies, S. Nguyen and E. Joyce for whole chromosome Oligopaints, and J. Kassis, S. Nguyen, and members of the Lei Laboratory for critical reading of the manuscript. This work was funded by the Intramural Program of the National Institute of Diabetes and Digestive and Kidney

Diseases, National Institutes of Health (DK015602 to E. P. L.). The funders had no role in study design, data collection and analysis, decision to publish, or preparation of the manuscript.

## Author contributions
Conceptualization: I. B., E. P. L.; methodology: I. B.; mass-spec: S. C.; Oligopaints: L. F. R.; software: Y. C., C.-Y. L., G.-Y. Y.; bioinformatics analysis: I. B., S. C.; validation: I. B., E. P. L.; formal analysis: I. B., E. P. L.; investigation: I. B.; data curation: C.-Y. L.; writing – original draft: I. B., E. P. L.; supervision: E. P. L.; funding acquisition: E. P. L.

## Funding

## Competing interests
The authors declare no competing interests.
