## [Peer Review File · Nature Communications]

REVIEWER COMMENTS

Reviewer #1 (Remarks to the Author):

Understanding the relationship between genome architecture and transcription is an important challenge. Evidence suggests that chromatin domain insulators play a role in this relationship. This manuscript reports an interesting interaction between the housekeeping gene activator M1BP and the chromatin domain insulator cofactor CP190, and provides strong evidence that this interaction is important for activation of many M1BP-associated genes. The authors attempt to link this to genome architecture. The data in the manuscript are of high quality and of interest to researchers interested in gene regulation and nuclear organization. However, the view presented in the Introduction and the analysis of the data could be improved. The authors should provide more balanced background and be careful not to overstate their results. Specifics follow, many of which are minor points.

1. Introduction, lines 45-55: The authors should point out differences between mammalian and *Drosophila* TAD boundaries. Hi-C evidence indicates that stable pairing between TAD boundaries (mediated by CTCF in mammals) is rarely seen in *Drosophila*. Also in contrast to mammals, several papers indicate that constitutively active genes form many *Drosophila* TAD boundaries. Maybe this active chromatin has short-range, local interactions that prevent longer range interactions between adjacent TADs. This suggests a central role for transcription in *Drosophila* TAD boundary formation. This is an important point that is consistent with the correlation between M1BP/CP190, transcription and TAD boundaries reported by the authors in their Results.

2. Introduction, line 69: More is known about insulator bodies than the authors claim. Published information indicates that insulator bodies are storage depots for insulator proteins, not aggregates of active insulators. Their formation is promoted by certain stresses and by SUMOylation. The authors should include this information.

3. Introduction, lines 106-110: It is fair to say that M1BP and CP190 could affect insulator activity and nuclear organization. But the effect of M1BP knockdown on insulator bodies could be a stress response, and the effect on local genome compaction could be due to fusion of adjacent TADs when transcription separating the TADs is decreased. These alternative interpretations should be mentioned.

4. Results, lines 181-183 and Supp. Fig. S2B: The authors make a point of saying their 5 day M1BP RNAi did not greatly alter cell viability, in contrast to a previous study. The authors should also point out that their results are consistent with the original M1BP paper from the Gilmour lab, which reported that effects on cell viability were not observed until after 5 days of M1BP knockdown (the study the authors cite used a 7 day knockdown). I know of another lab with a similar observation, although it was not published. Yet Supplemental Figure S2B shows that cell viability of the control knockdown started to be affected by day 5 while M1BP was not affected. This is strange, why was the control affected? Is it possible that the control and M1BP RNAi are mislabeled? Mislabeling would be consistent with the evidence that M1BP knockdown does affect cell viability, but not until after 5 or more days.

5. Results, lines 194-197: The sentence "Consistent with..." is unclear, it needs to be rewritten.

6. Results, line 245: 'important' rather than 'required' for expression? Expression is still detected after knockdown, so 'required' is an overstatement. This holds for lines 246 and 257 of this section.

7. Results, line 260: 'contributes to' or 'is important for' rather than 'is required for'? Similar to comment (5) above. Same for line 274: 'CP190-facilitated', not 'CP190-dependent' since reduced M1BP binding is detected at many affected sites after CP190 knockdown.

8. Results, line 279: This paragraph again overstates the results. M1BP binding is sensitive to but not completely dependent on CP190; genes require CP190 for full expression, but not for expression; there is a CP190-dependent reduction, but not loss, of M1BP binding.

9. Results, line 287: Chromatin association of CP190 'at some sites' is highly... Another overstatement of the results. This is also true of the last paragraph of this section (line 301 etc.). There are additional overstatements in the manuscript, I will leave it to the authors to find and correct them.

10. Results section starting at line 310: This section seems to make a circular argument. M1BP and CP190 are enriched at promoters near TAD boundaries. The authors have shown that M1BP and CP190 activate promoters (and that around 80% of M1BP peaks also have CP190 peaks; around 30% of CP190 peaks also have M1BP peaks). So it is not surprising that genes activated by M1BP and/or CP190 are close to TAD boundaries. Maybe the authors are trying to make some other point that is not clear. A point of interest not mentioned by the authors is that, as mentioned in point (1), there is evidence that transcription of constitutive genes defines many TAD boundaries in *Drosophila*. So M1BP and CP190 could play a role in defining these boundaries. This is a point the authors should make.

11. Results section starting at line 341: As mentioned in point (2), insulator bodies are not aggregates of active insulators, and their formation is promoted by certain stresses. Driving M1BP[RNAi] with Act5C-GAL4 is larval lethal, presumably because M1BP is important for housekeeping gene expression but it takes time to deplete maternal stores. Maybe the observed increase in insulator bodies per nucleus is due to stress caused by M1BP depletion. The authors should mention this possibility. Maybe the authors can measure the insulator body volume per nucleus using Z-stacks to determine if insulator bodies only fragment or also increase in volume after M1BP knockdown. In Fig 7A it looks like the volume increases.

12. Results section starting at line 353: If transcription forms some TAD boundaries, then decreasing transcription at a TAD boundary could allow the adjacent TADs to merge, as mentioned in point (1). This could explain the result obtained after M1BP knockdown, and why longer range effects were not observed. If CP190 knockdown also affects transcription in the two boundary regions examined, and also decreases M1BP binding, it is not clear why a similar effect was not seen. Still, this is something the authors should consider.

13. Discussion line 387: Is M1BP 'required for proper nuclear localization of insulator bodies', or does lack of M1BP cause stress that leads to formation of more insulator bodies?

14. Discussion section starting at line 391: Data for the role of M1BP at the Gypsy insulator are convincing, but: M1BP and Su(Hw) seem to mainly co-localize at Gypsy, not other Su(Hw) or M1BP sites; and many genomic Su(Hw) sites seem to be repressors rather than insulators. The authors did not mention the repressor activity of Su(Hw). Around 5% of M1BP colocalizes with Su(Hw); lower for Su(Hw) [has more sites than M1BP]. Yet this is how the authors found the relationship between M1BP and CP190 that they focus on. The authors should explain why they think the relationship between M1BP and Su(Hw) is so limited. This is a glaring omission. Might it be related to the repressor versus insulator functions of Su(Hw)?

15. Discussion lines 476-478: It looks like M1BP knockdown allows adjacent TADs to merge. Point (11) provides a potential explanation that is also consistent with the observed lack of longer range effects. I raised this point about TAD boundaries several times in my comments, indicating that there are several places where this concept could be included.

16. Discussion lines 440-441: This sentence should be modified to say that the role of CP190 in transcription is 'in part' rather than 'likely' through interactions with M1BP. The next sentence states that CP190 preferentially associates with promoters genome-wide. According to this manuscript, only 31% of CP190 peaks are near M1BP peaks and only 319 of 1382 genes downregulated after CP190 knockdown are also downregulated after M1BP knockdown. Transcriptional effects of CP190 must include collaboration with proteins in addition to M1BP.

17. Fig. 3E legend, line 899: How many TSSs are in the heatmaps? Is each heatmap independently sorted, or is the M1BP sorting used for all?

18. Fig. 7B legend, line 974: Histograms of insulator body volumes per nucleus would be useful.

Reviewer #2 (Remarks to the Author):

The manuscript by Bag et al. describes a new functional player of the gypsy insulator complex, the transcriptional regulator M1BP. Additionally, the manuscript explores the function of the insulator protein CP190 at promoters, as opposed to its more canonical role at chromatin insulators. The authors find that M1BP physically interacts with insulator proteins, and co-localizes with specifically CP190 at a subset of CP190-bound promoters that harbor Motif 1 consensus and that are often located at TAD borders. Importantly, the authors also find that MB1P is involved in recruitment of CP190 to such promoters, and that the two factors co-regulate active transcription at shared promoters. Depletion of M1BP is seen to "alter compaction" of certain chromatin regions, and together, the authors conclude that M1BP co-functions with CP190 to drive transcription at TADs.

Overall, the findings are significant and novel, and the data is very clean and of exceptional quality. The story is somewhat underdeveloped in mechanistic insight, particularly in relation to understanding how this transcriptional role of Cp190, and of M1BP, relates to architectural functions. These issues are detailed further below, under major comments.

Major comments:

1) The connection between this function of CP190 and M1BP as transcriptional regulators and integrity/function of TAD boundaries is unclear. Namely, does CP190/M1BP-dependent transcription at TAD boundaries contribute to TAD organization or inter-communication? I think regardless of the outcome of this inquiry, it is an important question to address: are transcriptional and architectural functions of these proteins related or independent? The paper is significant in either scenario, but seems incomplete without at least trying to address this question. The authors could address this by further imaging assays, involving oligopaints, such as looking at whether depletion of M1BP leads to more contact between neighboring TADs, as opposed to within the same TAD.

2) A related mechanistic question is what does M1BP-dependent "local compaction", reported in Figure 8, really represent? If this is chromatin/nucleosomal compaction, can the authors test it by a local or genome-wide assay that reports chromatin accessibility (MNase-qPCR, ATAC-seq, etc), in M1BP KD cells, at the regions affected by M1BP KD, such as shown in Figure 8? Alternatively, this compaction could represent an intermingling of previously separate TADs/loss of TAD borders, as discussed in point 1 – which should be addressed by the authors more explicitly than is currently present in the manuscript.

Minor comments:

1) line 50, "CTCF indeed is enriched at TAD borders and is required for TAD formation" needs a reference.

2) line 250, "expressed on a co-transfected control plasmid. (Fig. 5A, 5B)." - remove period after "plasmid".

3) It would be helpful to validate that MBP1 and Cp190 bind to the Motif 1-luciferase construct by ChIP-qPCR, and not to the mutant one, shown in Figure 5.

Reviewer #3 (Remarks to the Author):

The authors of the MS provide evidence that M1BP physically interacts with the complex of proteins associated with the insulator gypsy. Furthermore, using functional tests they demonstrate that M1BP plays a certain a role in the enhancer-blocking and barrier activity of this insulator. Additionally, the authors found that M1BP, Su(Hw), Mod(mdg4), and CP190 were associated with Su(Hw)-binding sites of gypsy

retrotransposon in KC cells and that depletion of either M1BP or CP190 resulted in a drastic loss of association of all core factors with gypsy. Finally, depletion of M1BP resulted in a spatial redistribution of gypsy within the nuclear volume.

In the other part of the work the authors demonstrate that M1BP preferentially colocalizes with CP190 genome-wide especially on M1 motif-containing promoters, but not with the other core components of gypsy insulator - Su(Hw) and Mod(mdg4). They also found that M1BP was not deposited on best-characterized endogenous insulator sequences (1A-2, Fab-7, Fab-8). Using depletion of either M1BP or CP190 the authors demonstrate that these proteins are involved in regulation of similar sets of genes across the genome. Cooperation of M1BP and CP190 in the activation of transcription from M1BP motif-containing promoters was confirmed by experiments with transient transfection of constructs harboring a reporter gene controlled by M1BP motif-containing promoters. The authors also present an evidence that (i) CP190 contributes somehow to the recruitment of M1BP to chromatin and vice versa, (ii) genes that require M1BP or CP190 for their transcription are located relatively close to TAD borders.

Overall, the manuscript looks rather descriptive. Many observations presented in the MS are confirmative rather than novel. Different parts of the study are not logically connected. Consequently, it is difficult to see what is the main message of the MS. Reading this article, I could not get rid of the impression that two independent works (the study of the role of M1BP in the activity of the gypsy insulator and the study of the cooperation of M1BP and CP190 in the activation of transcription of a subset of genes) were artificially combined into one article. The sections of the MS focused on TAD borders and chromatin do not contribute much to the understanding of the possible role of M1BP in higher order chromatin organization. The authors should generate Hi-C maps for cells with depletion of CP190 and M1BP, annotate TADs and chromatin compartments and compare their profiles with TADs and compartment profiles in control cells.

Besides these general considerations I have several technical comments.

1. It looks that experiments of coimmunoprecipitation of proteins associated with Cp190 and Su(HW) were performed in a single biological replicate that is not sufficient for making reliable conclusions, especially quantitative estimates. I am surprised by the fact that a protein which was directly fished by antibodies (Su(Hw) or CP190) is present in immunoprecipitates in a similar amounts comparing to associated proteins. Might it happen that large protein aggregates were precipitated? To this end, I do not think that in was good idea to use Polycomb protein as a negative control. It would be much better to use some component of active chromatin that was not reported to interact with gypsy insulator.
2. The authors should show in the supplement the original blots from which the strips shown in the figures were cut
3. lines 252-258 - Fig.5C. Could the authors explain why the expression of luciferase from constructs with a mutation of the M1BP binding site is reduced in cells with M1BP depletion.
4. lines 322-334. How does qPCR-mediated M1BP binding to gypsy correlates with genome-wide data? Is it possible to discriminate individual gypsy insertions on genome-wide ChIP-seq profiles.
5. Figure S5. Depletion of Su(Hw) is rather inefficient. Could this have affected the interpretation of the results of the comparison of luciferase activity.
6. lines 465-466. Although the authors state that CP190 is specifically associated with TAD borders, this specificity is rather limited, because, in Drosophila, all proteins of active chromatin are about equally overrepresented at TAD borders.
8. The original data of qPCR analysis should be presented in the Supplement.

We appreciate the reviewers' thoughtful comments and enthusiasm for our work. We have addressed each of their comments below. We have added three sets of Oligopaint FISH results specifically examining whether local compaction in M1BP-depleted cells is specific to TAD borders or also occurs within TADs. We found that both inter-TAD and intra-TAD compaction is increased in M1BP-depleted cells. Furthermore, we have added ATAC-seq data and analyses that demonstrate that M1BP but not CP190 promotes chromatin accessibility at M1BP binding sites, including transcriptionally activated genes and regions near TAD borders. As a result, the manuscript is improved with respect to mechanistic insight into this newly discovered role for M1BP in chromatin insulator function and genome organization.

Per journal requirements, we eliminated subheadings in the Discussion and reduced the length of subheadings in the Results to a maximum of 60 characters.

REVIEWER COMMENTS

Reviewer #1:

Understanding the relationship between genome architecture and transcription is an important challenge. Evidence suggests that chromatin domain insulators play a role in this relationship. This manuscript reports an interesting interaction between the housekeeping gene activator M1BP and the chromatin domain insulator cofactor CP190, and provides strong evidence that this interaction is important for activation of many M1BP-associated genes. The authors attempt to link this to genome architecture. The data in the manuscript are of high quality and of interest to researchers interested in gene regulation and nuclear organization. However, the view presented in the Introduction and the analysis of the data could be improved. The authors should provide more balanced background and be careful not to overstate their results. Specifics follow, many of which are minor points.

1. Introduction, lines 45-55: The authors should point out differences between mammalian and *Drosophila* TAD boundaries. Hi-C evidence indicates that stable pairing between TAD boundaries (mediated by CTCF in mammals) is rarely seen in *Drosophila*. Also in contrast to mammals, several papers indicate that constitutively active genes form many *Drosophila* TAD boundaries. Maybe this active chromatin has short-range, local interactions that prevent longer range interactions between adjacent TADs. This suggests a central role for transcription in *Drosophila* TAD boundary formation. This is an important point that is consistent with the correlation between M1BP/CP190, transcription and TAD boundaries reported by the authors in their Results.

We have added information to the Introduction about the central role of transcription in genome organization in *Drosophila* as well as the presence of constitutively active genes at TAD borders on p. 2.

2. Introduction, line 69: More is known about insulator bodies than the authors claim. Published information indicates that insulator bodies are storage depots for insulator proteins, not

aggregates of active insulators. Their formation is promoted by certain stresses and by SUMOylation. The authors should include this information.

We have added this information and appropriate references to the Introduction on p. 3.

3. Introduction, lines 106-110: It is fair to say that M1BP and CP190 could affect insulator activity and nuclear organization. But the effect of M1BP knockdown on insulator bodies could be a stress response, and the effect on local genome compaction could be due to fusion of adjacent TADs when transcription separating the TADs is decreased. These alternative interpretations should be mentioned.

We have added to the Discussion on p. 15 the possibility that depletion of M1BP results in cellular stress that promotes insulator body formation.

As suggested by Reviewer 2, we performed additional Oligopaint assays to address the possibility that adjacent TADs undergo fusion in M1BP-depleted cells. As our probes are limited to ~30 kb and the average size of active TADs is 22 kb, we were limited to the study of larger TADs, which fall primarily into lowly transcribed classes (inactive, Polycomb, and null). We analyzed three independent regions each corresponding to three 32 kb probes spaced 15 kb apart, two of which are located in the same TAD with the third located in the adjacent TAD. We found that in M1BP- but not CP190-depleted cells that both inter-TAD and intra-TAD compaction increases. We present these new results on p. 13 and in new Fig. 8 and also discuss the findings in the Discussion on pp. 17-18.

4. Results, lines 181-183 and Supp. Fig. S2B: The authors make a point of saying their 5 day M1BP RNAi did not greatly alter cell viability, in contrast to a previous study. The authors should also point out that their results are consistent with the original M1BP paper from the Gilmour lab, which reported that effects on cell viability were not observed until after 5 days of M1BP knockdown (the study the authors cite used a 7 day knockdown). I know of another lab with a similar observation, although it was not published. Yet Supplemental Figure S2B shows that cell viability of the control knockdown started to be affected by day 5 while M1BP was not affected. This is strange, why was the control affected? Is it possible that the control and M1BP RNAi are mislabeled? Mislabeled would be consistent with the evidence that M1BP knockdown does affect cell viability, but not until after 5 or more days.

We have added the Li and Gilmour reference on p. 7. Also, the reviewer is correct that we inadvertently reversed the labels in Supplemental Fig. 2B and have now corrected them.

5. Results, lines 194-197: The sentence “Consistent with...” is unclear, it needs to be rewritten.

Thank you for catching this error. We have made the suggested change.

6. Results, line 245: 'important' rather than 'required' for expression? Expression is still detected after knockdown, so 'required' is an overstatement. This holds for lines 246 and 257 of this section.

We have made the suggested changes.

7. Results, line 260: 'contributes to' or 'is important for' rather than 'is required for'? Similar to comment (5) above. Same for line 274: 'CP190-facilitated', not 'CP190-dependent' since reduced M1BP binding is detected at many affected sites after CP190 knockdown.

We have made the suggested changes.

8. Results, line 279: This paragraph again overstates the results. M1BP binding is sensitive to but not completely dependent on CP190; genes require CP190 for full expression, but not for expression; there is a CP190-dependent reduction, but not loss, of M1BP binding.

We have made the suggested changes.

9. Results, line 287: Chromatin association of CP190 'at some sites' is highly... Another overstatement of the results. This is also true of the last paragraph of this section (line 301 etc.). There are additional overstatements in the manuscript, I will leave it to the authors to find and correct them.

We have made the suggested changes throughout the manuscript.

10. Results section starting at line 310: This section seems to make a circular argument. M1BP and CP190 are enriched at promoters near TAD boundaries. The authors have shown that M1BP and CP190 activate promoters (and that around 80% of M1BP peaks also have CP190 peaks; around 30% of CP190 peaks also have M1BP peaks). So it is not surprising that genes activated by M1BP and/or CP190 are close to TAD boundaries. Maybe the authors are trying to make some other point that is not clear. A point of interest not mentioned by the authors is that, as mentioned in point (1), there is evidence that transcription of constitutive genes defines many TAD boundaries in *Drosophila*. So M1BP and CP190 could play a role in defining these boundaries. This is a point the authors should make.

We changed the wording to "wanted to verify" on p. 11. Furthermore, we reworded the statement in the discussion "Overall, our results provide evidence that M1BP and CP190 at TAD borders and perhaps their ability to activate transcription of constitutively expressed genes located at TAD borders, combined with the capacity of M1BP to promote chromatin accessibility, may play a role in genome organization." on p. 18.

11. Results section starting at line 341: As mentioned in point (2), insulator bodies are not aggregates of active insulators, and their formation is promoted by certain stresses. Driving M1BP[RNAi] with Act5C-GAL4 is larval lethal, presumably because M1BP is important for

housekeeping gene expression but it takes time to deplete maternal stores. Maybe the observed increase in insulator bodies per nucleus is due to stress caused by M1BP depletion. The authors should mention this possibility. Maybe the authors can measure the insulator body volume per nucleus using Z-stacks to determine if insulator bodies only fragment or also increase in volume after M1BP knockdown. In Fig 7A it looks like the volume increases.

We have added to the Discussion on p. 15 the possibility that depletion of M1BP results in cellular stress that promotes insulator body formation. Furthermore, images were taken of a single plane, so we added measurements of insulator body area to Fig. 7C. Average insulator body area decreases in M1BP knockdown flies. Below, we also show that total insulator body area per nucleus in the eye also decreases whether or not normalized to the nuclear area. We performed an unpaired t test, $p < 0.001$.

12. Results section starting at line 353: If transcription forms some TAD boundaries, then decreasing transcription at a TAD boundary could allow the adjacent TADs to merge, as mentioned in point (1). This could explain the result obtained after M1BP knockdown, and why longer range effects were not observed. If CP190 knockdown also affects transcription in the two boundary regions examined, and also decreases M1BP binding, it is not clear why a similar effect was not seen. Still, this is something the authors should consider.

As suggested by Reviewer 2, we performed additional Oligopaint assays to address the possibility that adjacent TADs undergo fusion in M1BP-depleted cells. We also performed ATAC-seq and found that M1BP but not CP190 promotes chromatin accessibility at M1BP binding sites, including transcriptionally activated genes and regions near TAD borders. Although CP190 and M1BP do affect transcription of similar sets of genes, they are not identical. We mention the possibility that effects on transcription are unrelated to changes in local compaction. We also discuss the possibility that both M1BP and CP190 are required for

transcription at TAD borders and that M1BP is additionally needed to promote chromatin accessibility in order to achieve proper genome organization on p. 18.

13. Discussion line 387: Is M1BP 'required for proper nuclear localization of insulator bodies', or does lack of M1BP cause stress that leads to formation of more insulator bodies?

We have added to the Discussion on p. 15 the possibility that depletion of M1BP results in cellular stress that promotes insulator body formation.

14. Discussion section starting at line 391: Data for the role of M1BP at the Gypsy insulator are convincing, but: M1BP and Su(Hw) seem to mainly co-localize at Gypsy, not other Su(Hw) or M1BP sites; and many genomic Su(Hw) sites seem to be repressors rather than insulators. The authors did not mention the repressor activity of Su(Hw). Around 5% of M1BP colocalizes with Su(Hw); lower for Su(Hw) [has more sites than M1BP]. Yet this is how the authors found the relationship between M1BP and CP190 that they focus on. The authors should explain why they think the relationship between M1BP and Su(Hw) is so limited. This is a glaring omission. Might it be related to the repressor versus insulator functions of Su(Hw)?

We have added to the Discussion on p. 15 that co-immunoprecipitation of M1BP and *gypsy* core proteins is substoichiometric and could reflect interaction off of chromatin. Furthermore, we noted that some Su(Hw)-binding sites have been observed to correspond to transcriptional repression activity, perhaps dependent on the presence or absence of particular interactors (Schwartz et al., 2012).

15. Discussion lines 476-478: It looks like M1BP knockdown allows adjacent TADs to merge. Point (11) provides a potential explanation that is also consistent with the observed lack of longer range effects. I raised this point about TAD boundaries several times in my comments, indicating that there are several places where this concept could be included.

As suggested by Reviewer 2, we performed additional Oligopaint assays to address the possibility that adjacent TADs undergo fusion in M1BP-depleted cells. We added discussion of these results on pp. 17-18.

16. Discussion lines 440-441: This sentence should be modified to say that the role of CP190 in transcription is 'in part' rather than 'likely' through interactions with M1BP. The next sentence states that CP190 preferentially associates with promoters genome-wide. According to this manuscript, only 31% of CP190 peaks are near M1BP peaks and only 319 of 1382 genes downregulated after CP190 knockdown are also downregulated after M1BP knockdown. Transcriptional effects of CP190 must include collaboration with proteins in addition to M1BP.

We have made the suggested change.

17. Fig. 3E legend, line 899: How many TSSs are in the heatmaps? Is each heatmap independently sorted, or is the M1BP sorting used for all?

We show 6307 TSS that are Motif 1-enriched in the heatmaps (listed in Supplementary Table 5). Each heatmap is independently sorted. We updated the Fig. 3E legend with this information.

18. Fig. 7B legend, line 974: Histograms of insulator body volumes per nucleus would be useful.

We added area measurements as requested to Fig. 7C.

Reviewer #2:

The manuscript by Bag et al. describes a new functional player of the gypsy insulator complex, the transcriptional regulator M1BP. Additionally, the manuscript explores the function of the insulator protein CP190 at promoters, as opposed to its more canonical role at chromatin insulators. The authors find that M1BP physically interacts with insulator proteins, and co-localizes with specifically CP190 at a subset of CP190-bound promoters that harbor Motif 1 consensus and that are often located at TAD borders. Importantly, the authors also find that M1BP is involved in recruitment of CP190 to such promoters, and that the two factors co-regulate active transcription at shared promoters. Depletion of M1BP is seen to “alter compaction” of certain chromatin regions, and together, the authors conclude that M1BP co-functions with CP190 to drive transcription at TADs.

Overall, the findings are significant and novel, and the data is very clean and of exceptional quality. The story is somewhat underdeveloped in mechanistic insight, particularly in relation to understanding how this transcriptional role of Cp190, and of M1BP, relates to architectural functions. These issues are detailed further below, under major comments.

Major comments:

1) The connection between this function of CP190 and M1BP as transcriptional regulators and integrity/function of TAD boundaries is unclear. Namely, does CP190/M1BP-dependent transcription at TAD boundaries contribute to TAD organization or inter-communication? I think regardless of the outcome of this inquiry, it is an important question to address: are transcriptional and architectural functions of these proteins related or independent? The paper is significant in either scenario, but seems incomplete without at least trying to address this question. The authors could address this by further imaging assays, involving oligopaints, such as looking at whether depletion of M1BP leads to more contact between neighboring TADs, as opposed to within the same TAD.

As suggested, we performed additional Oligopaint assays to address the possibility that adjacent TADs undergo fusion in M1BP-depleted cells. As our probes are limited to ~30 kb and the average size of active TADs is 22 kb, we were limited to the study of larger TADs, which fall primarily into lowly transcribed classes (inactive, Polycomb, and null). We analyzed three independent regions each corresponding to three 32 kb probes spaced 15 kb apart, two of which are located in the same TAD with the third located in the adjacent TAD. We found that in

M1BP- but not CP190-depleted cells that both inter-TAD and intra-TAD compaction increases. We present these new results on p. 13 and in new Fig. 8 and also discuss the findings in the Discussion on pp. 17-18.

2) A related mechanistic question is what does M1BP-dependent “local compaction”, reported in Figure 8, really represent? If this is chromatin/nucleosomal compaction, can the authors test it by a local or genome-wide assay that reports chromatin accessibility (MNase-qPCR, ATAC-seq, etc), in M1BP KD cells, at the regions affected by M1BP KD, such as shown in Figure 8? Alternatively, this compaction could represent an intermingling of previously separate TADs/loss of TAD borders, as discussed in point 1 – which should be addressed by the authors more explicitly than is currently present in the manuscript.

As suggested, we performed Assay for Transposase-Accessible Chromatin (ATAC-seq) in mock treated, M1BP-depleted and CP190-depleted Kc cells. We found substantial overall loss of genome-wide chromatin accessibility in M1BP-depleted cells but not in CP190-depleted cells. Almost all M1BP binding sites overlap with accessible sites. We found that M1BP promotes local chromatin accessibility at its binding sites, including transcriptionally activated genes and TAD borders. These findings correlate with increases in local compaction observed by Oligopaint FISH in M1BP- but not CP190-depleted cells. We reported the results on pp. 13-14, new Fig. 9, and Supplementary Fig 11. We also show ATAC-seq signals and differential peak calls corresponding to Oligopaint FISH probe regions in Fig. 8 and Supplementary Figs. 8 and 9.

Minor comments:

1) line 50, “CTCF indeed is enriched at TAD borders and is required for TAD formation” needs a reference.

We are limited to seventy references so have referenced two recent reviews at the beginning of this paragraph and emphasized that point by adding “reviewed in” on p. 2.

2) line 250, “expressed on a co-transfected control plasmid. (Fig. 5A, 5B).” - remove period after “plasmid”.

Thank you for catching this error.

3) It would be helpful to validate that MBP1 and Cp190 bind to the Motif 1-luciferase construct by ChIP-qPCR, and not to the mutant one, shown in Figure 5.

As requested, we performed ChIP-qPCR for M1BP and CP190 after plasmid transfection as in Fig. 5. We found that both M1BP and CP190 associate with the *RpL30* promoter but not the downstream luciferase reporter or plasmid origin sequence in wildtype but not mutant plasmids. We observed considerably lower levels of association for both M1BP and CP190 with this Motif 1-containing region compared to endogenous genome sequences likely because plasmid

constructs are known not to form chromatin efficiently. We were unable to recover RpLP1 promoter sequence in our chromatin immunoprecipitations above preimmune background levels despite the fact that M1BP has been shown to bind to both the RpLP1 and RpL30 wildtype but not mutant promoters *in vitro* in immobilized template assays (Baumann and Gilmour, 2017). We have included the data below:

Reviewer #3:

The authors of the MS provide evidence that M1BP physically interacts with the complex of proteins associated with the insulator *gypsy*. Furthermore, using functional tests they demonstrate that M1BP plays a certain a role in the enhancer-blocking and barrier activity of this insulator. Additionally, the authors found that M1BP, Su(Hw), Mod(mdg4), and CP190 were associated with Su(Hw)-binding sites of *gypsy* retrotransposon in KC cells and that depletion of either M1BP or CP190 resulted in a drastic loss of association of all core factors with *gypsy*. Finally, depletion of M1BP resulted in a spatial redistribution of *gypsy* within the nuclear volume.

In the other part of the work the authors demonstrate that M1BP preferentially colocalizes with CP190 genome-wide especially on M1 motif-containing promoters, but not with the other core components of *gypsy* insulator - Su(Hw) and Mod(mdg4). They also found that M1BP was not deposited on best-characterized endogenous insulator sequences (1A-2, Fab-7, Fab-8). Using depletion of either M1BP or CP190 the authors demonstrate that these proteins are involved in regulation of similar sets of genes across the genome. Cooperation of M1BP and CP190 in the activation of transcription from M1BP motif-containing promoters was confirmed by experiments with transient transfection of constructs harboring a reporter gene controlled by M1BP motif-containing promoters. The authors also present an evidence that (i) CP190 contributes somehow to the recruitment of M1BP to chromatin and vice versa, (ii) genes that require M1BP or CP190 for their transcription are located relatively close to TAD borders.

Overall, the manuscript looks rather descriptive. Many observations presented in the MS are confirmative rather than novel. Different parts of the study are not logically connected. Consequently, it is difficult to see what is the main message of the MS. Reading this article, I could not get rid of the impression that two independent works (the study of the role of M1BP in the activity of the *gypsy* insulator and the study of the cooperation of M1BP and CP190 in the activation of transcription of a subset of genes) were artificially combined into one article. The sections of the MS focused on TAD borders and chromatin do not contribute much to the understanding of the possible role of M1BP in higher order chromatin organization. The authors should generate Hi-C maps for cells with depletion of CP190 and M1BP, annotate TADs and chromatin compartments and compare their profiles with TADs and compartment profiles in control cells.

We appreciate the reviewer's perspective regarding two particular findings highlighted in the manuscript, that (1) M1BP promotes *gypsy* insulator activity and (2) M1BP cooperates with CP190 to activate transcription near TAD borders. These findings are logically connected because this is the first report that M1BP physically and functionally interacts with CP190, and CP190 is a universal insulator protein. CP190 is a core component of the *gypsy* insulator complex yet it also localizes to TAD borders, where it is predicted to have insulator activity. The presumption in the field is that TAD borders are actual insulator sites; however, this has only been shown for a small number of such sites. Therefore, it is important to include data showing that M1BP actually promotes CP190-dependent insulator activity.

As the reviewer suggested, we have generated Hi-C maps in control Kc cells or depleted for M1BP or CP190. Our read depth (minimum 137 M valid pairs) was lower than other recent high resolution studies (>200 M), and we have been able to examine the data at 2 kb resolution. We clearly observe no change in A/B compartments, and these profiles match previous studies well, so we are confident of this conclusion. We do observe very mild differences in TAD borders upon either knockdown; however, TAD border calls are highly variable depending on choice of normalization algorithm, methods used to calculate insulation, and many additional parameters. In fact, like others, we have had difficulty obtaining TAD borders that match previous studies well, and our overall conclusion is that these mild differences may not be biologically meaningful. Thus, we have elected not to include these Hi-C results in the current manuscript, which already includes nine main figures and 11 supplemental figures. As has been observed for either mammalian or *Drosophila* CTCF (Nora et al., 2017 and Kaushal et al., 2021), it is likely necessary to fully deplete (and perhaps very rapidly given cell cycle effects) either M1BP or CP190 in order to convincingly detect changes by Hi-C. As an alternative, we have included additional Oligopaint FISH data in new Fig. 8 to address potential fusion of TAD borders.

Besides these general considerations I have several technical comments.

1. It looks that experiments of coimmunoprecipitation of proteins associated with Cp190 and Su(Hw) were performed in a single biological replicate that is not sufficient for making reliable conclusions, especially quantitative estimates. I am surprised by the fact that a protein which was directly fished by antibodies (Su(Hw) or CP190) is present in immunoprecipitates in a similar amounts comparing to associated proteins. Might it happen that large protein aggregates were precipitated? To this end, I do not think that it was good idea to use Polycomb protein as a negative control. It would be much better to use some component of active chromatin that was not reported to interact with gypsy insulator.

For the quantitative mass spec analysis, 6 independent replicates were pooled together for analysis of both anti-Su(Hw) and anti-CP190 immunoprecipitates (noted in Materials and Methods on p. 21) as directed by the NIDDK mass spec facility director as best practice. Top 50 lists for mass spec results shown in Supplementary Table 1 and Supplementary Table 2 verify the specificity of the immunoprecipitations, including previously identified interactors, such as Su(Hw), CP190, Mod(mdg4), Pita, Ibf1/2, BEAF-32, CTCF, AGO2, HIP1, and CLAMP. We would also like to point out that also identified in both immunoprecipitations is Gzf, a known M1BP interactor (Baumann et al., 2018). Coimmunoprecipitation of anti-Su(Hw), anti-CP190, anti-Mod(mdg4)67.2, and anti-M1BP followed by Western blotting experiments were performed two times for each antibody, and a single experiment is shown (noted in Fig. 2 legend).

2. The authors should show in the supplement the original blots from which the strips shown in the figures were cut

We submitted original blots in additional Supplementary Data files.

3. lines 252-258 - Fig.5C. Could the authors explain why the expression of luciferase from constructs with a mutation of the M1BP binding site is reduced in cells with M1BP depletion.

It is not clear why depletion of M1BP also decreases luciferase expression of constructs with mutated Motif 1 relative to control plasmid Renilla expression. David Gilmour communicated to us that during their study (Baumann et al., 2018) they examined deletion constructs and identified upstream sequences that also contribute to expression but did not analyze these sequences in more detail. It is possible that these sequences are also sensitive to loss of M1BP.

4. lines 322-334. How does qPCR-mediated M1BP binding to gypsy correlates with genome-wide data? Is it possible to discriminate individual gypsy insertions on genome-wide ChIP-seq profiles.

In Fig. 6C, we provided the actual % input of chromatin immunoprecipitated with M1BP for 11 individual binding sites. In Fig. 6F, we provided the same measurement for Su(Hw)-binding sites within *gypsy*, and these levels are comparable. Because *gypsy* elements are multicopy across the genome, the level measured for these sites is an average across all sites amplified. We did not analyze repetitive sequence in the ChIP-seq data because of the complexity entailed for such an analysis, which would essentially derive the same information as the directed qPCR.

5. Figure S5. Depletion of Su(Hw) is rather inefficient. Could this have affected the interpretation of the results of the comparison of luciferase activity.

The reviewer is correct that we only observe approximately 60% of Su(Hw) depletion, but we do not expect Su(Hw) to affect Motif 1-dependent transcription because Su(Hw) does not generally associate with Motif 1 containing gene promoters or colocalize extensively with M1BP throughout the genome. Furthermore, efficient depletion of Mod(mdg4)^{67.2} also has no effect in this assay.

6. lines 465-466. Although the authors state that CP190 is specifically associated with TAD borders, this specificity is rather limited, because, in *Drosophila*, all proteins of active chromatin are about equally overrepresented at TAD borders.

The first Hi-C studies in *Drosophila* (Hou et al., 2012 and Sexton et al., 2012) identified CP190 as one of the factors most enriched at TAD borders. In our study, we use the highest resolution TAD border definitions available, which were defined in Kc cells (Ramirez et al., 2018). This study also verified that CP190 is highly enriched at TAD borders. These studies disagree with the reviewer's statement that all proteins associated with active chromatin are about equally overrepresented at TAD borders.

8. The original data of qPCR analysis should be presented in the Supplement.

We provided all Ct values for standard curves and replicates for all tested sites for ChIP-qPCR in additional Supplementary Data files.

REVIEWER COMMENTS

Reviewer #1 (Remarks to the Author):

Most of my concerns have been adequately addressed. In addition, there is substantial new data exploring effects of knockdowns on chromatin compaction in more detail. I have only two small comments and one more complicated concern about insulator bodies:

1) p. 11, line 312: "M1BP directly facilitates", not "M1BP is directly facilitates"

2) Is reference 10 correct, "Gradients of Intercellular CO₂ Levels Across the Leaf Mesophyll"?

3) Fig. 7c; p. 12 line 355; p. 37 line 1087: Insulator bodies are a minor point in the manuscript, although they are mentioned many times in various places (7 paragraphs and one figure). They are used as one piece of evidence for a role of M1BP in nuclear organization. Since single plane images (rather than z-stacks) of insulator bodies were available the authors took my suggestion and measured the area (rather than volume) of insulator bodies with and without M1BP knockdown, added as Fig. 7c. From the legend, line 1087, and the y-axis labels in Fig. 7c it is not clear what area is plotted. Is it area per insulator body, per nucleus, or something else? This should be clarified. What is relevant for my point below is area (or volume) per nucleus.

Another point about Fig. 7: It is not clear why none of the plots included in the response to reviewer's comments are the same as in Fig. 7c. Isn't the same data from eye cells being used? This suggests an inconsistency in data processing. One of the plots in the response to reviewer's comments shows that insulator area per nucleus decreases in eye cells after M1BP RNAi. My rough measurements of the outlined nuclei in Fig. 7a found the opposite, that the area increases around 4-fold. I also found increased area for the other cell types included in Fig. 7a. This is consistent with the increased numbers of insulator bodies per nucleus with only a small decrease in insulator body diameter. The increased insulator body volume per nucleus is a point I tried to raise in my original review, and is relevant to my point below.

Continuing with my concern about changes to insulator bodies after M1BP knockdown: The authors discuss changes to insulator bodies after M1BP knockdown as if M1BP is directly responsible. But insulator body formation is promoted by certain stresses. Knockdown of M1BP by RNAi causes stress, as evidenced by eventual effects on the cell cycle in cultured cells and larval lethality in animals. I raised the concern that changes to insulator bodies could be an indirect, stress-related effect. In this case, the volume of insulator bodies per nucleus would be expected to increase after M1BP knockdown (of course the same could be observed for direct M1BP involvement, for instance if M1BP inhibits insulator body formation). The larger number of smaller insulator bodies could be due to more nucleation sites induced by stress, while stress could cause the total volume per nucleus to increase as appears to be the case in Fig. 7a. The authors dealt with this concern in a very superficial manner. For instance, at line 355 the authors could mention that the increased number of insulator bodies per nucleus results in an increase in insulator volume per nucleus and could be due to stress effects indirectly related to M1BP knockdown.

To summarize my concerns about insulator bodies: The authors claim the change reflects a role of M1BP in insulator body formation, and by extension a role in nuclear organization. An alternative explanation is that it is an indirect, stress-related effect. Consistent with stress inducing insulator body formation, Fig. 7a appears to show an increase in insulator body area per nucleus after M1BP RNAi (although the authors claim the opposite in their response to reviewer's comments). This is why I wanted area per nucleus to be shown in Fig. 7c; I think Fig. 7c needs to be redone. It is not clear what is being shown in Fig. 7c, the legend (line 1087; p. 37) and y-axes need to be clearer. My rough analysis of Fig. 7a indicates the insulator body area per nucleus increases after M1BP knockdown, which is not shown in Fig. 7c. Additionally, the text should clearly state that the M1BP knockdown effect on insulator bodies could be indirect, so do not provide clear support for a direct role of M1BP in insulator body formation and nuclear organization. At a minimum this should be done around line 355 (p. 12). This is a minor point that does not have great impact on the manuscript, and only minor revisions are necessary to address this concern.

Reviewer #2 (Remarks to the Author):

The manuscript by Bag et al. describes a newly identified interplay between the transcriptional regulator M1BP and the insulator protein CP190 in transcription and genome architecture of TAD borders. In their revised manuscript, the authors have addressed all of my concerns and included new experiments such as FISH oligopaints assays and ATAC-seq to determine effects of M1BP and CP190 on TAD intermingling and chromatin compaction. The new data brings important mechanistic insights to their conclusions, and the manuscript is ready for publication in my opinion.

Reviewer #3 (Remarks to the Author):

In the revised version of the article, the authors shifted the emphasis towards the model postulating that active transcription in border regions promotes TAD spatial isolation, while the role of canonical insulator proteins in TAD insulation in *Drosophila* is not obvious. From the point of view of the reviewer, this shift in emphasis has improved the article. Indeed, all the results presented in the article can be explained within the framework of the model postulating that the localization of actively transcribed genes at the boundaries of TADs is necessary and sufficient for insulation of TADs. In this regard, however, the need to present data on the Gypsy insulator in this article raises even more questions. Indeed, a number of authors have shown that this insulator is not enriched at the TAD boundaries in *Drosophila*. In the text of the article, the authors provide convincing evidence that there is a certain cooperation in the binding of M1BP and CP190 to the genome and their participation in the activation of transcription. At the same time, M1BP and CP190 depletions have different effects on chromatin condensation. This apparent contradiction should be addressed in Discussion.

Specific comments

1. Introduction – “CP190 is highly enriched at TAD borders, suggesting a central role in TAD formation.” As mentioned in my comments to the original version of the MS, the fact that CP190 is highly enriched at TAD borders does not imply that CP190 has a central role in TAD formation. A number of proteins belonging to active chromatin, including RNA Pol II and nucleosome remodeling factors are equally enriched at TAD borders (see, for example, Rowley et al., 2017, *Mol Cell*; Ulianov et al., 2016, *Genome Res*). The fact that depletion of either CP190 or M1BP performed by authors did not affect TAD profiles questions the central role of CP190 at TAD borders. In my opinion these results should be presented in a supplementary figure.
2. Introduction – “General inhibition of transcription using chemical treatments or heat shock results in acute disruption of TADs and compartments, but the mechanistic details of how transcription contributes to genome organization are yet to be elucidated”. This statement is not supported by any references and is likely incorrect. The only two related publications I can recall are Barutcu et al., 2019, *EMBO Rep* and Li et al., 2015, *Mol Cell*. In both cases the effect of transcription arrest on TAD and compartment profiles was rather moderate. In no case it could be qualified as “acute disruption”
3. In relation to my original comment 5. If the authors did not expect *Su(Hw)* to affect Motif 1-dependent transcription, why at all they performed depletion of *Su(Hw)*. In any case 60% depletion does not permit to make any conclusion. It is better to remove this result from the MS.

We appreciate the reviewers' enthusiasm for our work and the opportunity to further revise the manuscript. We have addressed each of their comments below.

REVIEWER COMMENTS

Reviewer #1:

Most of my concerns have been adequately addressed. In addition, there is substantial new data exploring effects of knockdowns on chromatin compaction in more detail. I have only two small comments and one more complicated concern about insulator bodies:

1) p. 11, line 312: "M1BP directly facilitates", not "M1BP is directly facilitates"

We have corrected this typo.

2) Is reference 10 correct, "Gradients of Intercellular CO(2) Levels Across the Leaf Mesophyll"?

Thank you for pointing this out. We have corrected this reference (now reference 7).

3) Fig. 7c; p. 12 line 355; p. 37 line 1087: Insulator bodies are a minor point in the manuscript, although they are mentioned many times in various places (7 paragraphs and one figure). They are used as one piece of evidence for a role of M1BP in nuclear organization. Since single plane images (rather than z-stacks) of insulator bodies were available the authors took my suggestion and measured the area (rather than volume) of insulator bodies with and without M1BP knockdown, added as Fig. 7c. From the legend, line 1087, and the y-axis labels in Fig. 7c it is not clear what area is plotted. Is it area per insulator body, per nucleus, or something else? This should be clarified. What is relevant for my point below is area (or volume) per nucleus.

We plotted area per insulator body in Fig. 7c and have clarified the legend. As requested, we added total insulator body area per nucleus to Fig. 7d and corresponding text on p. 12.

Another point about Fig. 7: It is not clear why none of the plots included in the response to reviewer's comments are the same as in Fig. 7c. Isn't the same data from eye cells being used? This suggests an inconsistency in data processing. One of the plots in the response to reviewer's comments shows that insulator area per nucleus decreases in eye cells after M1BP RNAi. My rough measurements of the outlined nuclei in Fig. 7a found the opposite, that the area increases around 4-fold. I also found increased area for the other cell types included in Fig. 7a. This is consistent with the increased numbers of insulator bodies per nucleus with only a small decrease in insulator body diameter. The increased insulator body volume per nucleus is a point I tried to raise in my original review, and is relevant to my point below.

In the original response to reviewers, we misunderstood the reviewer's request and had calculated area of one randomly selected insulator body per nucleus, nuclear area, and insulator body area normalized to nuclear area.

Continuing with my concern about changes to insulator bodies after M1BP knockdown: The authors discuss changes to insulator bodies after M1BP knockdown as if M1BP is directly responsible. But insulator body formation is promoted by certain stresses. Knockdown of M1BP by RNAi causes stress, as evidenced by eventual effects on the cell cycle in cultured cells and larval lethality in animals. I raised the concern that changes to insulator bodies could be an indirect, stress-related effect. In this case, the volume of insulator bodies per nucleus would be expected to increase after M1BP knockdown (of course the same could be observed for direct M1BP involvement, for instance if M1BP inhibits insulator body formation). The larger number of smaller insulator bodies could be due to more nucleation sites induced by stress, while stress could cause the total volume per nucleus to increase as appears to be the case in Fig. 7a. The authors dealt with this concern in a very superficial manner. For instance, at line 355 the authors could mention that the increased number of insulator bodies per nucleus results in an increase in insulator volume per nucleus and could be due to stress effects indirectly related to M1BP knockdown.

As requested, we added total insulator body area per nucleus to Fig. 7d and corresponding text on p. 12. The reviewer is correct that total insulator body area per nucleus increases in M1BP-depleted cells. We already mentioned in the Discussion on p. 15 that changes in insulator body localization could be due to stress in M1BP-depleted cells.

To summarize my concerns about insulator bodies: The authors claim the change reflects a role of M1BP in insulator body formation, and by extension a role in nuclear organization. An alternative explanation is that it is an indirect, stress-related effect. Consistent with stress inducing insulator body formation, Fig. 7a appears to show an increase in insulator body area per nucleus after M1BP RNAi (although the authors claim the opposite in their response to reviewer's comments). This is why I wanted area per nucleus to be shown in Fig. 7c; I think Fig. 7c needs to be redone. It is not clear what is being shown in Fig. 7c, the legend (line 1087; p. 37) and y-axes need to be clearer. My rough analysis of Fig. 7a indicates the insulator body area per nucleus increases after M1BP knockdown, which is not shown in Fig. 7c. Additionally, the text should clearly state that the M1BP knockdown effect on insulator bodies could be indirect, so do not provide clear support for a direct role of M1BP in insulator body formation and nuclear organization. At a minimum this should be done around line 355 (p. 12). This is a minor point that does not have great impact on the manuscript, and only minor revisions are necessary to address this concern.

We never state anywhere in the manuscript that the effect of M1BP depletion on insulator bodies is direct. We modified text on p. 12 and p. 14 to reduce the likelihood of such interpretation. We already stated on p. 15 of the Discussion that depletion of M1BP could result in cellular stress that induces insulator body formation. It is customary to present data in the Results section and interpretation of results in the Discussion section.

Reviewer #2:

The manuscript by Bag et al. describes a newly identified interplay between the transcriptional regulator M1BP and the insulator protein CP190 in transcription and genome architecture of TAD borders. In their revised manuscript, the authors have addressed all of my concerns and included new experiments such as FISH oligopaints assays and ATAC-seq to determine effects of M1BP and CP190 on TAD intermingling and chromatin compaction. The new data brings important mechanistic insights to their conclusions, and the manuscript is ready for publication in my opinion.

We appreciate the constructive comments previously suggested by this reviewer.

Reviewer #3:

In the revised version of the article, the authors shifted the emphasis towards the model postulating that active transcription in border regions promotes TAD spatial isolation, while the role of canonical insulator proteins in TAD insulation in *Drosophila* is not obvious. From the point of view of the reviewer, this shift in emphasis has improved the article. Indeed, all the results presented in the article can be explained within the framework of the model postulating that the localization of actively transcribed genes at the boundaries of TADs is necessary and sufficient for insulation of TADs. In this regard, however, the need to present data on the Gypsy insulator in this article raises even more questions. Indeed, a number of authors have shown that this insulator is not enriched at the TAD boundaries in *Drosophila*.

In the text of the article, the authors provide convincing evidence that there is a certain cooperation in the binding of M1BP and CP190 to the genome and their participation in the activation of transcription. At the same time, M1BP and CP190 depletions have different effects on chromatin condensation. This apparent contradiction should be addressed in Discussion.

We increased the discussion regarding this apparent contradiction on p. 18.

Specific comments

1. Introduction – “CP190 is highly enriched at TAD borders, suggesting a central role in TAD formation.”

As mentioned in my comments to the original version of the MS, the fact that CP190 is highly enriched at TAD borders does not imply that CP190 has a central role in TAD formation. A number of proteins belonging to active chromatin, including RNA Pol II and nucleosome remodeling factors are equally enriched at TAD borders (see, for example, Rowley et al., 2017, *Mol Cell*; Ulianov et al., 2016, *Genome Res*). The fact that depletion of either CP190 or M1BP performed by authors did not affect TAD profiles questions the central role of CP190 at TAD borders. In my opinion these results should be presented in a supplementary figure.

We changed “central” to “possible” on p. 2 of the Introduction.

As mentioned previously, we have only been able to analyze the Hi-C data from a single replicate of cells for each sample, with less than ideal sequencing depth. As many manuscripts begin with in-depth Hi-C analysis, we are uncomfortable with publishing preliminary data of this nature as supplemental data. Because TADs are very small in *Drosophila* relative to mammals and border calls are highly variable depending on analysis parameters, we are hesitant to make any strong conclusions especially for cells bearing incomplete depletion of each factor.

2. Introduction – “General inhibition of transcription using chemical treatments or heat shock results in acute disruption of TADs and compartments, but the mechanistic details of how transcription contributes to genome organization are yet to be elucidated”.

This statement is not supported by any references and is likely incorrect. The only two related publications I can recall are Barutcu et al., 2019, EMBO Rep and Li et al., 2015, Mol Cell. In both cases the effect of transcription arrest on TAD and compartment profiles was rather moderate. In no case it could be qualified as “acute disruption”

We eliminated the word “acute” on p. 2, and Rowley et al., 2018; Hug et al., 2017; and Li et al., 2015 are already cited for the preceding related statement regarding transcription and genome organization.

3. In relation to my original comment 5.

If the authors did not expect Su(Hw) to affect Motif 1-dependent transcription, why at all they performed depletion of Su(Hw). In any case 60% depletion does not permit to make any conclusion. It is better to remove this result from the MS.

As requested, we have eliminated the data regarding Su(Hw)-depletion from Fig. 5 and Supplementary Fig. 5.